



# Co-occurrence of Fe and P stress in natural populations of the marine diazotroph *Trichodesmium*

Noelle A. Held[1,2], Eric A. Webb[3], Matthew M. McIlvin[1], David A. Hutchins[3], Natalie R. Cohen[1], Dawn M. Moran[1], Korinna Kunde[4], Maeve C. Lohan[4], Claire M. Mahaffey[5], E. Malcolm S. Woodward[6], Mak A. Saito[1*]

[1]Department of Marine Chemistry and Geochemistry, Woods Hole Oceanographic Institution, Woods Hole, MA 02543 USA
[2]Department of Earth, Atmospheric, and Planetary Sciences, Massachusetts Institute of Technology, Cambridge, MA. 02139 USA
[3]Marine and Environmental Biology, Department of Biological Sciences, University of Southern California, Los Angeles, CA, 90089 USA
[4]Ocean and Earth Science, National Oceanography Centre, University of Southampton, Southampton, UK
[5]Department of Earth, Ocean and Ecological Sciences, University of Liverpool, Liverpool, UK
[6]Plymouth Marine Laboratory, Plymouth, UK

*Correspondence to*: Mak Saito (msaito@whoi.edu)

**Abstract.** *Trichodesmium* is a globally important marine microbe that provides fixed nitrogen (N) to otherwise N limited ecosystems. In nature, nitrogen fixation is likely regulated by iron or phosphate availability, but the extent and interaction of these controls are unclear. From metaproteomics analyses using established protein biomarkers for iron and phosphate stress, we found that co-stress is the norm rather than the exception for field *Trichodesmium* colonies. Counter-intuitively, the nitrogenase enzyme was most abundant under co-stress, consistent with the idea that *Trichodesmium* has a specific physiological state under nutrient co-stress. Organic nitrogen uptake was observed to occur simultaneously with nitrogen fixation. Quantification of the phosphate ABC transporter PstC combined with a cellular model of nutrient uptake suggested that *Trichodesmium* is confronted by the biophysical limits of membrane space and diffusion rates for iron and phosphate acquisition. Colony formation may benefit nutrient acquisition from particulate and organic nutrient sources, alleviating these pressures. The results indicate that to predict the behavior of *Trichodesmium*, we must consider multiple nutrients simultaneously across biogeochemical contexts.

## 1. Introduction

The diazotrophic cyanobacterium *Trichodesmium* plays an important ecological and biogeochemical role in the tropical and subtropical oceans globally. By providing bioavailable nitrogen (N) to otherwise N-limited ecosystems, it supports basin-scale food webs, increasing primary productivity and carbon flux from the surface ocean (Capone, 1997; Carpenter and Romans, 1991; Coles et al., 2004; Deutsch et al., 2007; Sohm et al., 2011). Nitrogen fixation is energetically and nutritionally expensive, so it typically occurs when other sources of N are unavailable, i.e. in N-starved environments (Karl et al., 2002). However, nitrogen availability is not the sole control on nitrogen fixation, which must be balanced against





the cell's overall nutritional status. Because it can access a theoretically unlimited supply of atmospheric N$_2$, *Trichodesmium*

often becomes phosphorus (P) limited (Frischkorn et al., 2018; Hynes et al., 2009; Orchard, 2010; Sañudo-Wilhelmy et al., 2001; Wu et al., 2000). It also has a tendency to drive itself towards iron (Fe) limitation because the nitrogenase enzyme is iron-demanding (Bergman et al., 2013; Chappell et al., 2012; Rouco et al., 2018; Sunda, 2012; Walworth et al., 2016).

There is uncertainty about when and where *Trichodesmium* is Fe and P stressed and how this impacts nitrogen fixation in nature. Some reports suggest that *Trichodesmium* is primarily phosphate stressed in the North Atlantic, and

primarily Fe stressed in the Pacific, owing to relative Fe and P availability in these regions (Bergman et al., 2013; Chappell et al., 2012; Frischkorn et al., 2018; Hynes et al., 2009; Orchard, 2010; Sañudo-Wilhelmy et al., 2001). However, others have suggested that Fe and P can be co-limiting to *Trichodesmium*; one incubation study found two examples of Fe/P co-limitation in the field (Mills et al., 2004). Even less clear is how Fe and/or P stress impacts nitrogen fixation. For instance, despite the intuitive suggestion that nitrogen fixation is limited by Fe or P availability, laboratory evidence indicated that

*Trichodesmium* is specifically adapted to co-limited conditions, with higher growth and N$_2$-fixation rates under co-limitation than under single nutrient limitation (Garcia et al., 2015; Walworth et al., 2016).

There are several established protein biomarkers for Fe and P stress in *Trichodesmium*, all of which are periplasmic binding proteins involved in nutrient acquisition. For Fe, this includes the IdiA and IsiB proteins and for phosphorus, specifically phosphate, the PstS and SphX proteins (see Table S1). In *Trichodesmium*, IsiB, a flavodoxin, and IdiA, an ABC

transport protein, are expressed under Fe limiting conditions, and both are conserved across species with high sequence identity (Chappell et al., 2012; Webb et al., 2007). Transcriptomic and proteomic studies have shown that they are more abundant under Fe stress conditions (Chappell et al., 2012; Snow et al., 2015; Walworth et al., 2016). In this dataset, IsiB and IdiA were both highly abundant and correlated to one another (Figure S1). IdiA is used as the molecular biomarker of Fe stress in the following discussion, but the same conclusions could be drawn from IsiB distributions. Like IdiA and IsiB,

SphX and PstS are conserved across diverse *Trichodesmium* species (Chappell et al., 2012; Walworth et al., 2016). SphX is abundant at the transcript and protein level under phosphate limitation (Orchard et al., 2009; Orchard, 2010). PstS, a homologous protein located a few genes downstream of SphX, responds less clearly to phosphate stress. In *Trichodesmium*, the reason may be that PstS is not preceded by a Pho box, which is necessary for P based regulation (Orchard et al., 2009). Thus, in this study we focused on SphX as a measure of phosphate stress and IdiA as a marker of Fe stress.

Here, we present evidence based on field metaproteomes that *Trichodesmium* colonies were simultaneously Fe and P stressed throughout the world's oceans, but particularly in the tropical and subtropical Atlantic. While Fe/P stress has been suggested before, this study provides molecular evidence for co-stress in a broad geographical and temporal survey. This co-stress occurred across significant gradients in Fe and P concentration, suggesting nutrient stress was driven not only by biogeochemical gradients but also by *Trichodesmium's* inherent physiology; we explore possible biophysical and

biochemical mechanisms behind this. Fe and P stress were positively associated with nitrogen fixation and organic nitrogen uptake, suggesting that *Trichodesmium's* Fe, P, and N statuses are closely linked.



## 2. Materials and methods

### 2.1 Sample acquisition

A total of 37 samples were examined in this study. Samples were acquired by the authors on various research expeditions and most exist in biological duplicate or triplicate (Table S2). *Trichodesmium* colonies were hand-picked from 200 μm or 130 μm surface plankton net tows, rinsed thrice in 0.2 μm filtered trace metal clean surface seawater into trace metal clean LDPE bottles, decanted onto 0.2-5 μm filters, and frozen until protein extraction. The samples were of mixed puff and tuff morphology types, depending on the natural diversity present at the sampling location. The majority of samples considered in this study were taken in the early morning pre-dawn hours. Details such as filter size, morphology, location, cruise, date, and time of sampling are provided in Table S1.

### 2.2 Sample acquisition

Proteins were extracted by a detergent based method following Saito et al. (2014) and Lu et al. (2005). To reduce protein loss and contamination, all tubes were ethanol rinsed and dried prior to use and all water and organic solvents used were LC/MS grade. Sample filters were placed in a tube with 1-2 mL 1% sodium dodecyl sulfate (SDS) extraction buffer (1% SDS, 0.1 M Tris/HCL pH 7.5, 10 mM EDTA) and incubated for 10 min at 95℃ with shaking, then for one hour at room temperature with shaking. The protein extract was decanted and clarified by centrifugation (14100xg) at room temperature. The crude protein extracts were quantified with the colormetric BCA protein concentration assay with bovine serum albumin as a standard (Pierce catalog number 23225). Extracts were concentrated by 5 kD membrane centrifugation (Vivaspin spin columns, GE Healthcare). The protein extracts were purified by organic precipitation in 0.5 mM HCl made in 50% methanol and 50% acetone at -20 ℃ for at least one week, then collected by centrifugation at 14100xg for 30 min at 4 ℃, decanted and dried by vacuum concentration for 10min. The protein pellets were re-suspended in a minimum amount of 1% SDS extraction buffer, and re-quantified by BCA protein concentration assay to assess extraction efficiency.

The proteins were embedded in a 500 μL final volume acrylamide gel, which was then cut up into 1 mm pieces to maximize surface area and rinsed in 50:50 acetonitrile: 25 mM ammonium bicarbonate overnight at room temperature. The next morning, the rinse solution was replaced and the rinse repeated for 1 hour. Gels were dehydrated thrice in acetonitrile, dried by vacuum centrifugation, and rehydrated in 10 mM dithiothrietol (DTT) in 25 mM ammonium bicarbonate, then incubated for one hour at 56 ℃ with shaking. Unabsorbed DTT solution was removed and the volume recorded, allowing for calculation of the total gel volume. Gels were washed in 25 mM ammonium bicarbonate, then incubated in 55 mM iodacetamide for one hour at room temperature in the dark. Gels were again dehydrated thrice in acetonitrile. Trypsin (Promega Gold) was added at a ratio of 1:20 μg total protein in 25 mM ammonium bicarbonate in a volume sufficient to barely cover the gel pieces. Proteins were digested overnight at 37 ℃ with shaking. Any unabsorbed solution was then removed to a new tube and 50μL of peptide extraction buffer (50% acetonitrile, 5% formic acid in water) was added and incubated for 20 min at room temperature. The supernatant as then decanted and combined with the unabsorbed solution, and



the step then repeated. The resulting peptide mixture was concentrated by vacuum centrifugation to 1 µg µL⁻¹ concentration. Finally, the peptides were clarified by centrifugation at room temperature, taking the top 90% of the volume to reduce the carry over of gel debris.

### 2.3 Sample acquisition

The global proteomes were analyzed by online comprehensive active-modulation two-dimensional liquid chromatography (LC x LC-MS) using high and low pH reverse phase chromatography with inline PLRP-S (Agilent) and C18 columns packed in house. 10 ug of protein was injected per run directly onto the first column using a Thermo Dionex Ultimate3000 RSLCnano system (Waltham, MA), and an additional RSLCnano pump was used for the second dimension gradient. The samples were then analyzed on a Thermo Orbitrap Fusion mass spectrometer with a Thermo Flex ion source
(Waltham, MA).

### 2.4 Relative quantitation of peptides and proteins

Raw spectra were searched with the Sequest algorithm using a custom-built genomic database (Eng, Fischer, Grossmann, and MacCoss, 2008). The genomic database consisted of a publically available *Trichodesmium* community metagenome
available on the JGI IMG platform (IMG ID 2821474806), as well as the entire contents of the CyanoGEBA project genomes (Shih et al., 2013). Protein annotations were derived from the original metagenomes. SequestHT mass tolerances were set at +/- 10ppm (parent) and +/- 0.8 Dalton (fragment). Cysteine modification of +57.022 and methionine modification of +16 were included. Protein identifications were made with Peptide Prophet in Scaffold (Proteome Software) at the 95% protein and peptide identification levels. Relative abundance was measured as normalized top 3 precursor (MS1) intensities.
Normalization and false discovery rate (FDR) calculations, which were 0.1% peptide and 1.2% protein, were performed in Scaffold (Proteome Software). The mass spectrometry proteomics data have been deposited to the ProteomeXchange Consortium via the PRIDE partner repository with the dataset identifier PXD016225 and 10.6019/PXD016225 (Perez-Riverol et al., 2019). Statistical tests of relationships between proteins were conducted with the scipy stats package (https://docs.scipy.org/doc/scipy/reference/stats.html) using linear Pearson tests when the relationship appeared to be linear
and a Spearman rank order test when this was not the case.

### 2.5 Absolute quantitation of peptides

A small number of peptides were selected for absolute quantitation using a modified heterologous expression system. The peptides were ensured to be specific to *Trichodesmium* species based on sequence identity compared to over
300 marine bacteria genomes, three metagenomes, and 956 specialized assemblies (see www.metatryp.whoi.edu) (Saito et al., 2015). A custom plasmid was designed that contained the *Escherichia coli* K12 optimized reverse translation sequences for peptides of interest separated by tryptic spacers (protein sequence = TPELFR). To avoid repetition of the spacer nucleotide sequence, twelve different codons were utilized to encode the spacer. Six equine apomyoglobin and three





peptides from the commercially available Pierce peptide retention time calibration mixture (product number 88320) were also included. The sequence was inserted into a pet(30a)+ plasmid using the BAMH1 5' and XhoI 3' restriction sites.

The plasmid was transformed into competent tuner(DE3)pLys *E.coli* cells and grown on kanamycin amended LB agar plates to ensure plasmid incorporation. A single colony was used to inoculate a small amount of kanamycin containing $^{15}$N labelled S.O.C. media (Cambridge Isotope Laboratories) as a starter culture. These cells were grown overnight and then used to inoculate 10 mL of $^{15}$N labeled, kanamycin-containing SOC media. Cells were grown to approximately OD600 0.6,

then induced with 1 mM isopropyl β-D-1-thiogalactopyranoside (IPTG), incubated in the overexpression phase overnight at room temperature and harvested by centrifugation.

Cells were lysed with BugBuster detergent with added benzonase nuclease. The extracts were centrifuged and a large pellet of insoluble cellular material remained. Because the plasmid protein was large, this pellet contained a large number of inclusion bodies containing nearly pure protein. The inclusion bodies were solubilized in 6 M urea at 4 °C

overnight. The protein was reduced, alkylated, and trypsin digested in solution to generate a standard peptide mixture.

The standard mixture was calibrated to establish the exact concentration of the peptides. A known amount (10 fmol $\mu L^{-1}$) of the commercially available Pierce standard peptide mixture (Catalog number 88320) and an apomyoglobin digest was spiked into the standard. The ratio of Pierce (isotopically labelled according to JPT standards) or apomyoglobin (light) to heavy standard peptide MS2 peak area was calculated and used to establish the final concentration of the standard peptide

mixture (Fu et al., 2016; Milo, 2013). Multiple peptides were used for this calibration and the standard deviation among them was approximately 10%. Finally, the linearity of the peptide standard was tested by generating a dilution curve and ensuring that the concentration of each peptide versus MS2 peak area was linear between 0.001 and 20 fmol $\mu L^{-1}$ concentration.

The sample was prepared at 0.2 $\mu g$ $\mu L^{-1}$ concentration, with 10 $\mu L$ injected to give a total of 2 $\mu g$ total protein

analyzed. The heavy labelled standard peptide mixture was spiked into each sample at a concentration of 10 fmol $\mu L^{-1}$. The concentration of the light peptide was calculated as the ratio of the MS2 area of the light:heavy peptide multiplied by 10 $\mu g$ $\mu L^{-1}$. A correction was applied for protein recovery before and after purification, and the result was the absolute concentration of the peptide in fmol $\mu g^{-1}$ total protein.

The percent of the membrane occupied by the ABC transporter PstC was calculated by converting the absolute

protein concentration to molecules per *Trichodesmium* cell, using average values for *Trichodesmium* cell volume (Hynes et al., 2012), carbon content per volume (Strathmann, 1967), protein content per g carbon (Rouwenhorst, et al., 1991), and the cross sectional area of a calcium ATPase (Hudson et al., 1992) (see Table S3).

### 2.6 Self-organizing map analyses

Self-organizing maps were used to reduce the dimensionality of the data and explore relationships among co-varying proteins of interest. Only *Trichodesmium* proteins were considered. Analyses were conducted in Python 3.0 and fully reproducible code is available at https://github.com/naheld/self_organizing_map_tricho_metaP.





The input data consisted of a table of protein names (rows) and samples (columns) such that the input vectors contained 2818 features. To eliminate effects of scaling, the data was unit normalized with the Scikit-learn pre-processing algorithm. The input vectors were used to initialize a 100 output node (10x10) self-organizing map using the SOMPY Python library (https://github.com/sevamoo/SOMPY). The output nodes were then clustered using a k-means clustering algorithm (k = 10) implemented in scikit learn. The input nodes (proteins) assigned to each map node were then retrieved and the entire process repeated 10,000 times. Proteins were considered in the same cluster if they appeared in the same cluster of output nodes more than 99.99% of the time.

## 3. Results and Discussion

### 3.1 Proteome overview

This study presents 36 field metaproteomes of colonial *Trichodesmium* populations collected at sixteen locations on four expeditions (Table S2). Most samples were from the subtropical and tropical Atlantic and were collected in the early morning hours to avoid changes occurring on the diel cycle (Figure 1 and Table S2). At each location, *Trichodesmium* colonies were hand picked from plankton net tows, rinsed in filtered seawater, collected onto filters, and immediately frozen. The metaproteomes were analyzed with a two-dimensional LC-MS/MS workflow that provided deep coverage of the proteome. This resulted in 4478 protein identifications, of which 2944 were *Trichodesmium* proteins. The remaining proteins were from colony-associated epibionts, and will be discussed in a future publication. Protein abundance is presented as precursor (MS1) intensity of the three most abundant peptides for each protein, normalized to total protein in the sample. Thus, changes in protein abundance were interpreted as changes in the fraction of the proteome devoted to that protein. The most abundant were GroEL, ribosome, and phycobilisome proteins.

A self-organizing map analysis identified groups of proteins with similar profiles, i.e. proteins whose abundances changed cohesively, indicative of proteins that may be regulated similarly (Reddy et al., 2016). This revealed the central importance of nitrogen fixation to *Trichodesmium*. The nitrogenase proteins were among the most abundant in the proteome and were located in clusters 1 and 2 (Figure 2 and Table S3). Also in these clusters were nitrogen metabolism proteins including glutamine synthetase, glutamine hydrolyzing guanosine monophosphate (GMP) synthase and glutamate racemase. This is consistent with previous reports finding that N assimilation is synchronized with nitrogen fixation (Carpenter et al., 1992).

Nitrogen fixation was closely linked to carbon fixation. Many photosystem proteins clustered with the nitrogenase proteins, including phycobilisome proteins, photosystem proteins, and the citric acid cycle protein 2-oxoglutarate dehydrogenase. This clustering indicated direct regulatory links between C and N fixation. The nitrogen regulators P-II and NtcA were also present in this cluster and may mediate this association. In non-nitrogen fixing cyanobacteria, high abundance of the nitrogen regulators NtcA and P-II is indicative of nitrogen stress (Flores and Herrero, 2005; Saito et al., 2014). In diazotrophs, the role of these regulators is unclear because they do not respond to nitrogen compounds such as ammonia as they do in other cyanobacteria (Forchhammer and De Marsac, 1994). Here, clustering of NtcA and P-II with C





and N fixation proteins suggests that they play a role in balancing these processes in field populations, though the details of this role has yet to be elucidated.

In addition to identifying links between C and N fixation, the self-organizing map analysis demonstrated that field populations of *Trichodesmium* invest heavily in macro- and micro-nutrient acquisition. There were clusters of proteins

involved in trace metal acquisition and management, including Fe, zinc, and metal transport proteins, with the latter including proteins likely involved in Ni and Mo uptake (protein IDs TCCM_0270.00000020 and TCCM_0481.00000160). We also noted clusters of proteins involved in phosphate acquisition. Importantly, SphX and PstS appear in separate clusters, highlighting differential regulation of these functionally similar proteins.

**3.2 *Trichodesmium* is simultaneously iron and phosphate stressed throughout its habitat**

A surprising emergent observation from the *Trichodesmium* metaproteomes was the co-occurrence of multiple nutrient stress biomarkers in all samples. Biomarkers for iron (IdiA) and phosphate (SphX) stress were highly abundant and positively associated with surface Fe or P concentrations, following oceanographic trends (Figure 3). For instance, SphX varied up to 7.5 fold and was more abundant in populations from the North Atlantic gyre (JC150 expedition) compared with populations from near the Amazon river plume (Tricolim expedition) or at station ALOHA, where phosphate concentrations

were greater (see Figure S2) (Hynes et al., 2009; Sañudo-Wilhelmy et al., 2001; Wu et al., 2000). IdiA varied up to 8 fold, and increased moving West to East across the JC150 transect, consistent with an observed decrease in dFe concentrations (see Figure S2) (Kunde et al., 2019).

The ubiquitous and abundant presence of both Fe and P stress biomarkers implied that co-stress was the norm rather than the exception for *Trichodesmium* colonies in the field. This interpretation is supported by laboratory studies

demonstrating that IdiA and SphX are more abundant under Fe or P depletion, respectively (Chappell et al., 2012; Orchard, 2010; Snow et al., 2015; Walworth et al., 2016; Webb et al., 2007). This conclusion supports the growing acknowledgement that *Trichodesmium* experiences both Fe and phosphate stress throughout its habitat (Mills et al., 2004). If an implicit goal in understanding *Trichodesmium's* molecular physiology is to model the organism's behavior, these metaproteomes suggest that both Fe and phosphorus conditions must be jointly considered.

**3.3 The intersection of Fe, P and N stress**

The metaproteomes enabled the relationship between Fe and P stress and overall cellular metabolism to be explored. Nitrogenase protein abundance was positively correlated with both IdiA and SphX, and was in fact highest at the intersection of high Fe and P stress (Figure 4). This observation contrasts with the current paradigm that *Trichodesmium* down regulates nitrogen fixation when it is Fe or P stressed. Instead, it is consistent with the idea that the nutritional

demands of nitrogen fixation could drive the organism to Fe and P stress, thereby initiating an increase in Fe and P acquisition proteins including IdiA and SphX. This indicates that the cell's N, P and Fe statuses are linked, perhaps involving a regulatory network as is common in marine bacteria (Figure 5) (Held et al., 2019). This network may regulate a specific



physiological adaptation to nutrient co-stress. For instance, Fe and P co-limited *Trichodesmium* cells may reduce their cell size to optimize their surface area: volume quotient for nutrient uptake. However, a putative cell size biomarker Tery_1090,

while abundant in co-limited cells in culture, was not identified in these metaproteomes despite bioinformatic efforts to target it, likely because it is a low abundance protein (Walworth et al., 2016).

Nitrogen fixation is not the only way that *Trichodesmium* can acquire fixed N (Dyhrman et al., 2006; Küpper et al., 2008; Mills et al., 2004; Sañudo-Wilhelmy et al., 2001). In culture, *Trichodesmium* can be grown on multiple nitrogen sources including urea; in fact, it has been reported that nitrogen fixation provides less than 20% of the fixed N demand of

cells, and a revised nitrogen fixation model suggests that *Trichodesmium* takes up fixed nitrogen in the field (McGillicuddy , 2014; Mulholland and Capone, 1999). In this dataset, a urea ABC transporter was abundant, indicating that urea could be an important source of fixed nitrogen to colonies (Figure 6a). The transporter is unambiguously attributed to *Trichodesmium* rather than a member of the epibiont community. Of course, this does not rule out the possibility that urea or other organic nitrogen sources such as trimethylamine (TMA) are also utilized by epibionts, although no such epibiont transporters were

identified in the metaproteomes.

Typically, elevated urea concentration decreases or eliminates nitrogen fixation in colonies (Ohki, et al., 1991). However, in laboratory studies urea exposure must be unrealistically high (often over 20 μM) for this to occur, compared with natural concentrations which are much lower (Ohki et al., 1991; Wang et al., 2000). In the field, urea utilization and nitrogen fixation seem to occur simultaneously, with a urea uptake protein positively correlated to nitrogenase abundance

(Figure 6b). Urea and other organic nitrogen sources such as trimethylamine (TMA) could be sources of nitrogen for *Trichodesmium*, and the relationship to nitrogenase abundance may indicate a general N stress signature driving both organic nitrogen uptake and nitrogen fixation (Walworth et al., 2018). Alternatively, urea uptake could be a colony-specific behavior, since colonies were sampled here as opposed to laboratory cultures that typically grow as single filaments. For instance, urea could be used for recycling of fixed N within the colony, or there could be heterogeneity in nitrogen fixation,

with some cells taking up organic nitrogen and others fixing it. These unexpected observations of co-occurring nitrogen fixation and organic nitrogen transport show the value of exploratory metaproteomics, which does not require targeting of a specific protein based on a prior hypothesis.

### 3.4 Mechanisms of simultaneous iron and phosphate stress – membrane crowding

ABC transporters are multi-unit, trans-membrane protein complexes that use ATP to shuttle substrates across

membranes. Specific ABC transporters are required for iron versus phosphate uptake (Chappell et al., 2012; Orchard et al., 2009). Nutrient transport rates can be modulated by changing the number of uptake ligands installed on the cell membrane or the efficiency of the uptake ligands through expression of assisting proteins such as IdiA and SphX, which bind Fe or P respectively in the periplasm and shuttle the elements to their respective membrane transport complexes (Hudson and Morel, 1992). The high abundance of proteins involved in ABC transport suggested that nutrient transport rates could limit the





amount of Fe and P *Trichodesmium* can acquire. Thus, we explored whether membrane crowding, i.e. lack of membrane space, can constrain nutrient acquisition by *Trichodesmium*.

To investigate this, we quantified the absolute concentration of the phosphate ABC transporter PstC, which interacts with the phosphate stress biomarkers SphX and PstS. This analysis is distinct from the above global metaproteomes, which allowed patterns to be identified but did not allow for absolute quantitation of the proteins. The analysis was performed

similar to an isotope dilution experiment where labelled peptide standards are used to control for analytical biases. The analysis was performed for three Tricolim and six JC150 stations. Briefly, [15]N labelled peptide standards were prepared and spiked into the samples prior to PRM LC-MS/MS analysis. The concentration of the peptide in fmol μg[-1] total protein was calculated using the ratio of product ion intensities for the heavy (spike) and light (sample) peptide and converted to PstC molecules per cell (Table 1 and see also Table S4). The peptide used for quantitation of PstC was specific to *Trichodesmium*

species. Based on these calculations, on average 19 to 36% of the membrane was occupied by the PstC transporter. In one population (JC150 expedition, Station 7), up to 83% of the membrane was occupied by PstC alone. While these are first estimates, it is clear that the majority of *Trichodesmium* cells devoted a large fraction of their membrane surface area to phosphate uptake.

To examine whether membrane crowding can indeed cause nutrient stress or limitation, we developed a model of

cellular nutrient uptake in *Trichodesmium*. The model identifies the concentration at which free Fe or phosphate limits the growth of *Trichodesmium* cells. This is distinct from nutrient stress, which changes the cell's physiological state but does not necessarily impact growth. In the model, nutrient limitation occurs when the daily cellular requirement is greater than the uptake rate, a function of the cell's growth rate and elemental quota. Following the example of Hudson and Morel (1992), the model assumes that intake of nutrients once bound to the ABC transporter ligand is instantaneous, i.e. that nutrient

uptake is limited by formation of the metal-ligand complex at the cell surface. This is an idealized scenario, because if intake is the slow step, for instance in a high affinity transport system, the uptake rate would be slower and nutrient limitation exacerbated (discussed below).

We considered two types of nutrient limitation in the model (Table S5). First, we considered a diffusion-limited case, in which the rate of uptake is determined by diffusion of the nutrient to the cell's boundary layer ($\mu$*$Q = \frac{2}{3}k_D$[nutrient],

where $\mu$ = the cell growth rate, $Q$ = the cell nutrient quota, and $k_D$ = the diffusion rate constant, dependent on the surface area and diffusion coefficient of the nutrient in seawater). Based on empirical evidence provided by Hudson and Morel (1992), limitation occurs when the cell quota is greater than $\frac{2}{3}$ the diffusive-limited flux because beyond this, depletion of the nutrient in the boundary layer occurs. In the second case, membrane crowding limitation, the rate of uptake is determined by the rate of ligand-metal complex formation ($\mu$*$Q = k_f$[transport ligand][nutrient], where $k_f$ = the rate of ligand-nutrient

complex formation). Here, up to 50% of the membrane can be occupied by the transport ligand following the example of and Morel (1992). This is within the range of the above estimates of membrane occupation by phosphate transporter PstC. The model uses conservative estimates for diffusion coefficients, cell quotas, growth rates, and membrane space occupation to identify the lowest concentration threshold for nutrient limitation; as a result it is likely that *Trichodesmium* becomes limited





at higher nutrient concentrations than the model suggests. At this time, the model can only consider labile dissolved Fe and

inorganic phosphate, though *Trichodesmium* can also acquire particulate iron, organic phosphorus, phosphite, and

phosphonates (Dyhrman et al., 2006; Frischkorn et al., 2018; Polyviou et al., 2015; Poorvin et al., 2004; Rubin et al., 2011).

We first considered a spherical cell, where the surface area: volume quotient decreases as cell radius increases

(Figure 7). As the cell grows in size, higher nutrient concentrations are required to sustain growth. This is consistent with the

general understanding that larger microbial cells with lower surface area: volume quotient are less competitive in nutrient

uptake (Chisholm, 1992; Hudson and Morel, 1992). For a given surface area: volume quotient, we take the driver of nutrient

limitation to be whichever model (membrane crowding or diffusion limitation) requires higher nutrient concentrations to

sustain growth. For a spherical cell, Fe limitation is driven by diffusion when the cell is large and the surface area: volume

quotient is low (Figure 7a). However, when cells are smaller and the surface area: volume quotient is high, membrane

crowding drives nutrient limitation, meaning that the number of ligands, and not diffusion from the surrounding

environment, is the primary control on nutrient uptake. For phosphate, diffusion is almost always the driver of nutrient

limitation owing to the higher rate of ligand-nutrient complex formation ($k_f$) for phosphate, which causes very fast

membrane transport rates and relieves membrane-crowding pressures across all cell sizes (Figure 7b) (Froelich et al., 1982).

While this model may be directly applicable to some $N_2$-fixing cyanobacteria such as Groups B and C, which have

roughly spherical cells, *Trichodesmium* cells are not spheres but rather roughly cylindrical (Hynes et al., 2012). Thus, we

repeated the model calculations for cylinders with varying radii (r) and heights (2r or 10r) based on previous estimates of

*Trichodesmium* cell sizes (Bergman et al., 2013; Hynes et al., 2012). Cylinders have lower surface area: volume quotient

than spheres of similar sizes. In addition, the rate constant ($k_D$) for diffusion, which is a function of cell geometry, is greater.

This increases the slope of the diffusion limitation line such that membrane crowding is important across a greater range of

cell sizes (Figure 7c-d). *Trichodesmium* cell sizes vary in nature, for instance the cylinder height can be elongated,

improving the surface area: volume quotient. However, the impact of cell elongation to radius r and height 10r on both

diffusion limitation and membrane crowding is subtle (Figure 7e-f). Thus, we conclude that in certain scenarios, lack of

membrane space could indeed limit Fe and perhaps P acquisition by *Trichodesmium*.

A key assumption of the model is that uptake rates are instantaneous. In the above calculations, we use the

dissociation kinetics of Fe from water and phosphorus with common seawater cations as the best case (i.e. fastest possible)

kinetic scenario for nutrient acquisition. The model does not account for delays caused by internalization kinetics, which

would exacerbate nutrient limitation. For instance it does not consider nutrient speciation, which could affect internalization

rates, particularly for Fe (Hudson and Morel, 1992). Furthermore, the responsiveness of the periplasmic binding proteins

IdiA and SphX/PstS to environmental abundance (Figure 3) suggests that uptake is not simultaneous; their involvement is

likely associated with a kinetic rate of binding and dissociation from the periplasmic proteins in addition to any rate of ABC

transport.

Membrane crowding could produce real cellular challenges, leading to the observation of Fe and P co-stress across

the field populations examined. The above model explicitly allows 50% of the cell surface area to be occupied by any one





type of transporter, consistent with our estimate of cell surface area occupied by the PstC transporter. If 50% of the membrane is occupied by phosphate transporters, and another 50% for Fe transporters, this would leave no room for other
essential membrane proteins and even the membrane lipids themselves. The problem is further exacerbated if the cell installs transporters for nitrogen compounds such as urea, as the metaproteomes suggest. Thus, installation of transporters for any one nutrient must be balanced against transporters for other nutrients. This interpretation is inconsistent with Liebig's law of nutrient limitation, which assumes that nutrients are independent (Liebig, 1855; Saito et al., 2008). In an oligotrophic environment, membrane crowding could explicitly link cellular Fe, P, and N uptake status, driving the cell to be co-stressed
for multiple nutrients.

### 3.5 Advantages of the colonial form

Living in a colony has specific advantages and disadvantages for a *Trichodesmium* cell. Colonies may be able to access nutrient sources that would be infeasible for use by single cells or filaments. For instance, *Trichodesmium* colonies have a remarkable ability to entrain dust particles and can move these particles into the center of said colony (Basu et al.,
2019; Basu and Shaked, 2018; Poorvin et al., 2004; Rubin et al., 2011). In this study, which focused on *Trichodesmium* colonies, the chemotaxis response regulator CheY was very abundant, particularly in populations sampled near the Amazon and Orinoco river plumes. CheY was positively correlated with Fe stress biomarker IdiA, but not with phosphate stress biomarker SphX, suggesting that chemotactic movement is involved in entrainment of trace metals including from particulate sources (Figure 8).

The metaproteomes and nutrient uptake model presented in this paper support the growing understanding that *Trichodesmium* must be able to access particulate and organic matter. Living in a colony can be advantageous because such substrates can be concentrated, improving the viability of extracellular nutrient uptake systems. *Trichodesmium's* epibiont community produces siderophores, which assist in Fe uptake, particularly from particulate organic matter (Chappell and Webb, 2010; Lee et al., 2018). Siderophore production is energetically and nutritionally expensive, so it is most
advantageous when resource concentrations are high and loss is low, as would occur in the center of a colony (Leventhal et al., 2019). Colonies may similarly enjoy advantages for phosphate acquisition, particularly when the excreted enzyme alkaline phosphatase is utilized to access organic sources (Frischkorn et al., 2018; Elizabeth Duncan Orchard, 2010; Orcutt et al., 2013; Yamaguchi et al., 2016; Yentsch et al., 1970). Additionally, the concentration of cells in a colony means that the products of nitrogen fixation, including urea, can be recycled and are less likely to be lost to the environment. By increasing
effective size and concentrating deterrent toxins, colony formation may also protect against grazing (Hawser et al., 1992).

A key hallmark of *Trichodesmium* colony formation is production of mucus, which can capture particulate matter and concentrate it within the colony. In addition to particle entrainment, the mucus layer can benefit cells by protecting them from oxygen, facilitating epibiont associations, regulating buoyancy, defending against grazers and helping to "stick" trichomes together (Eichner et al., 2019; Lee et al., 2017; Sheridan, 2002). However, these benefits come at a cost because
the mucus layer hinders diffusion to the cell surface (Figure 9), reducing contact with the surrounding seawater. Despite this,



the benefits of colony formation seem to outweigh the costs, since *Trichodesmium* forms colonies in the field, particularly under stress (Bergman et al., 2013; Capone et al., 1997; Hynes et al., 2012).

**4. Conclusions**

*Trichodesmium*'s colonial lifestyle likely produces challenges for dissolved Fe and P acquisition, which must be compensated for by production of multiple nutrient transport systems, such as for particulate iron and organic phosphorous, at a considerable cost. While laboratory studies have largely focused on single nutrient stresses in free filaments, these metaproteomic observations and accompanying nutrient uptake model demonstrate that Fe and P co-stress is the norm rather than the exception. This means that the emphasis on single limiting nutrients in culture studies and biological models may not capture the complexities of *Trichodesmium's* physiology in situ. Thus, biogeochemical models should consider

incorporating Fe and P co-stress conditions. Specifically, in this study and in others there is evidence that nitrogen fixation is optimal under co-limited or co-stressed conditions, implying that an input of either Fe or P could counter-intuitively decrease $N_2$ driven new production (Garcia et al., 2015; Walworth et al., 2016).

These data demonstrate that *Trichodesmium* cells are confronted by the biophysical limits of membrane space and diffusion rates for their Fe, P, and possibly urea, acquisition systems. This means that there is little room available for

systems that interact with other resources such as light, $CO_2$, Ni, and other trace metals, providing a mechanism by which nutrient stress could compromise acquisition of other supplies. The cell membrane could be a key link allowing *Trichodesmium* to optimize its physiology in response to multiple environmental stimuli. This is particularly important in an ocean where nutrient availability is sporadic and unpredictable. Future studies should aim to characterize the specific regulatory systems, chemical species and phases, and symbiotic interactions that underlie *Trichodesmium's* unique behavior

and lifestyle.

**Data Availability**

All new data is provided in the supplementary material. The mass spectrometry proteomics data have been deposited to the ProteomeXchange Consortium via the PRIDE partner repository with the dataset identifier PXD016225 and

10.6019/PXD016225 (Perez-Riverol et al., 2019).

**Supplement**

Supplementary information is provided in a separate file (Figure S1, Table S1, Table S2, Table S6), with Tables S3, S4, and S5 provided separately due to their large sizes.


**Author Contributions**





N.A. and M.S. conceptualized the study. D.H. and E.W. lead the Tricolim expedition. C.M. and M.L. lead the JC150 expedition. N.C., M.W., and K.K. measured nutrient distributions on the Tricolim and JC150 expeditions. D.M. and M.M. helped with proteomics analyses. N.A. prepared the manuscript with contributions from all co-authors.


**Competing Interests**

The authors declare no competing interests.

**Acknowledgements**

We acknowledge Elena Cerdan Garcia, Asa Conover, Joanna Harley, Despo Polyviou, and Petroc Shelley for assistance with sampling and nutrient measurements while at sea, in addition to the entire crew of the JC150 and Tricolim expeditions. We thank Ben Van Mooy for insightful discussions regarding this work. This work was supported by a National Science Foundation Graduate Research Fellowship under grant 1122274 [N.Held], Gordon and Betty Moore Foundation grant number 3782 [M.Saito], and National Science Foundation grants OCE-1657755 and EarthCube-1639714 [M.Saito]. We also

acknowledge funding from the UK Natural Environment Research Council (NERC) grants awarded to CM (NE/N001079/1) and ML (NE/N001125/1). References

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






**Figures**

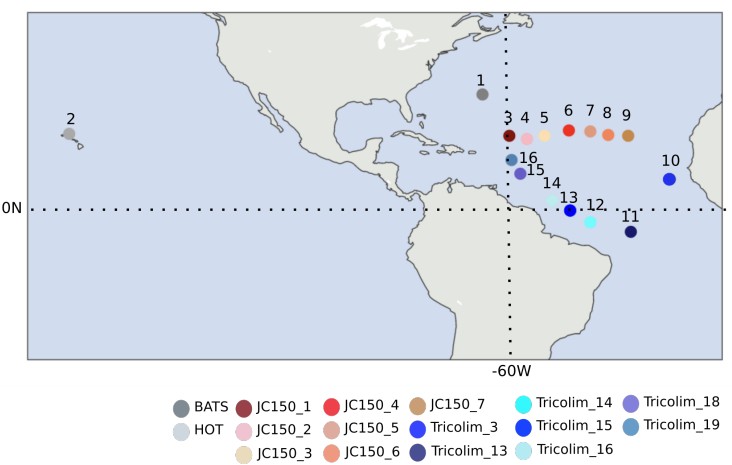

**Figure 1. Sampling locations. Red/pink colors indicate JC150 stations; blue colors indicate Tricolim stations, dark grey indicates the Bermuda Atlantic Time Series (BATS) and light grey indicates Hawaii Ocean Time Series (HOT). Most samples exist in duplicate or triplicate; see Table S2 for detailed information.**

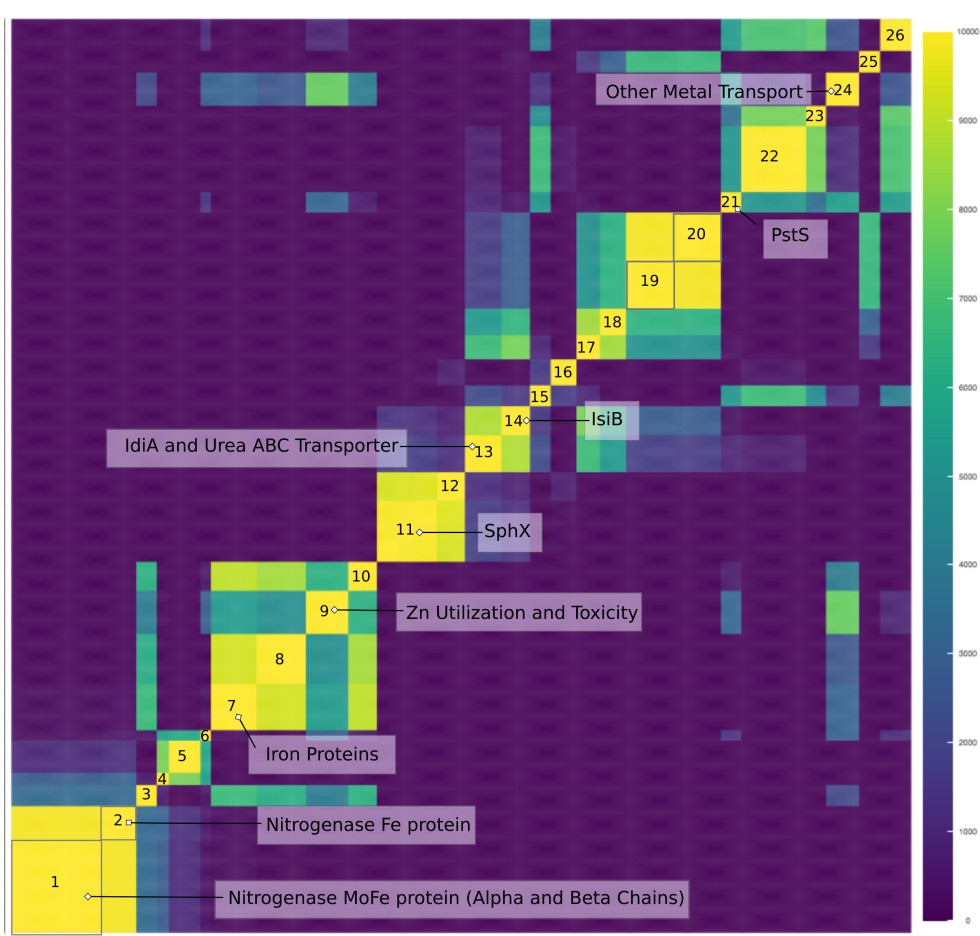

**Figure 2. Heatmap displaying results of self-organizing map analysis. Each protein was mapped to a self-organizing**
**map grid, and the grids subsequently clustered by a k-means clustering algorithm. The process was repeated 10,000**
**times and the results displayed here as a heatmap with warm colors representing proteins that appear in the same**
**cluster. Only the top 500 most abundant proteins are displayed. Dark yellow indicates proteins that appear in the**
**same cluster 99.99% of the time. Clusters # 1 and 2 contain nitrogen fixation, carbon fixation, and nitrogen**
**assimilation proteins as well as the regulatory systems NtcA and P-II. The cluster assignments for the proteins are**
**available in Table S4.**





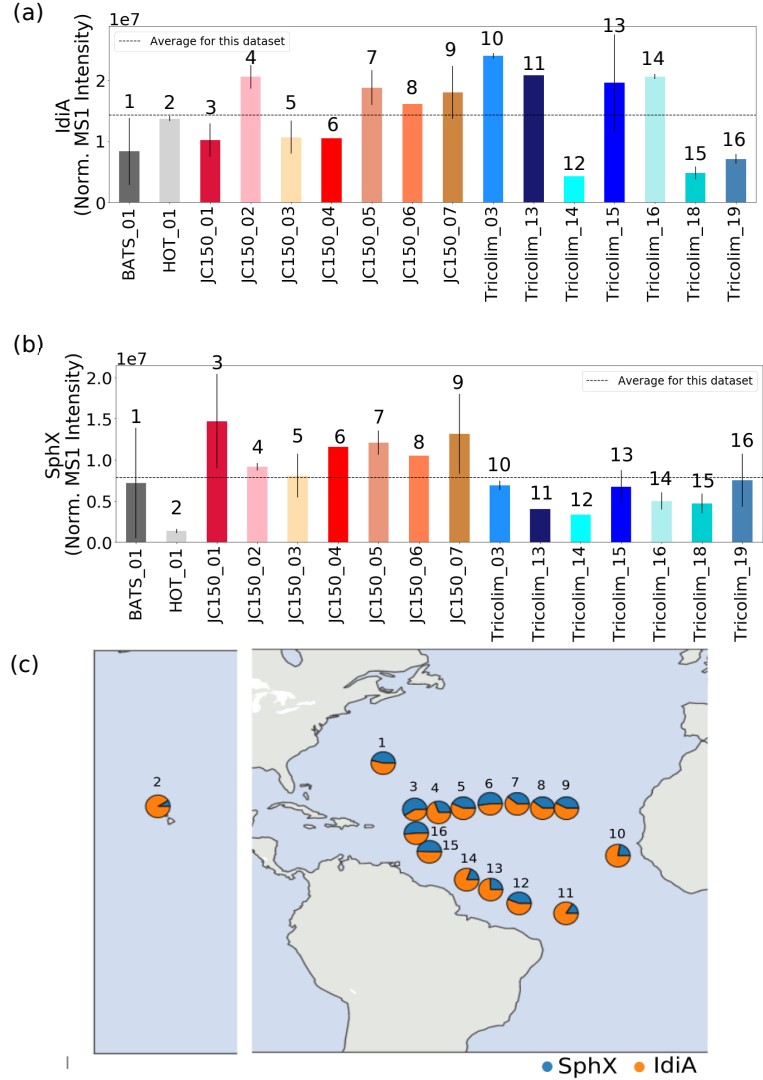

**Figure 3. (A) Relative abundance of iron stress protein IdiA (A) and phosphate stress protein SphX (B). IdiA and SphX were among the most abundant proteins in the entire dataset. Error bars are one standard deviation on the mean when multiple samples were available. Dashed lines represent average values across the dataset. (C) Relative abundance of IdiA (orange) and SphX (blue) overlaid on the sampling locations.**



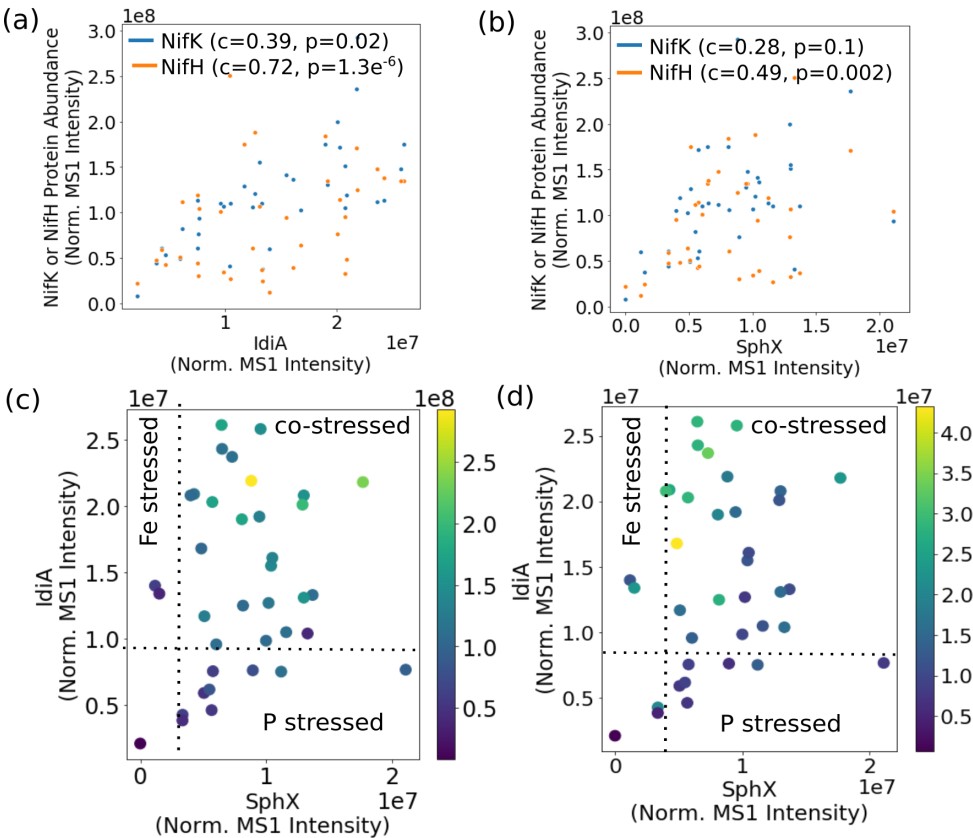

**Figure 4. Nitrogenase abundance is highest at the intersection of high iron and phosphate stress. A) IdiA and B) SphX abundance is positively related to nitrogenase MoFe and Fe protein abundance (c = Spearman rank-order correlation coefficient, p = Spearman p-value). Effects of combined iron and phosphate stress biomarkers on nitrogenase abundance. Marker colors represent abundance of NifK (panel C) and NifH (panel D).**





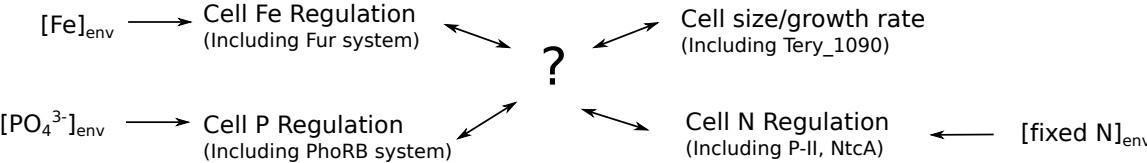

**Figure 5. The metaproteomes suggest that there is a currently unknown regulatory link between cellular Fe, P, and N**
**regulation. Key: Fur = ferric uptake regulator, PhoRB = phosphate two component sensory system, Tery_1090 =**
**putative cell size regulator, P-II/NtcA = nitrogen regulatory proteins.**

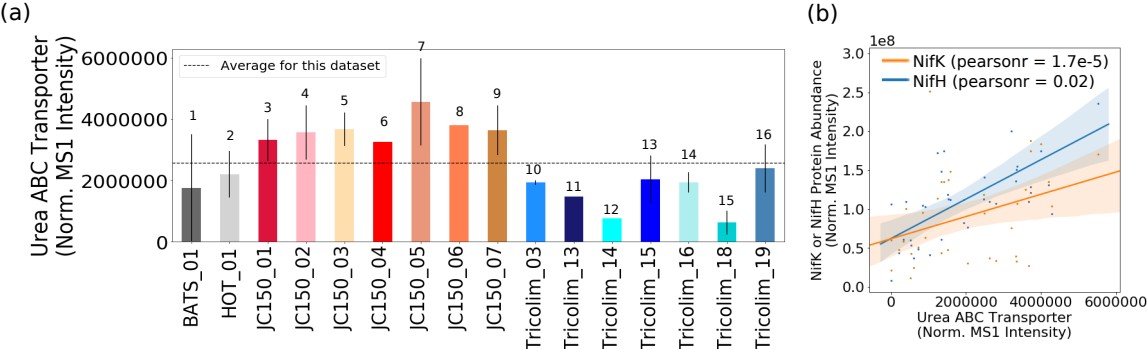


**Figure 6. A) Relative abundance of the *Trichodesmium* urea ABC transporter.  B) The abundance of the urea ABC**
**transporter is positively correlated with NifH and NifK abundance. Pearson linear correlation coefficients (r values)**
**are provided (p value for NifK = 1.7e$^{-5}$, NifH = 0.02). Shaded bars indicate 95% confidence intervals.**





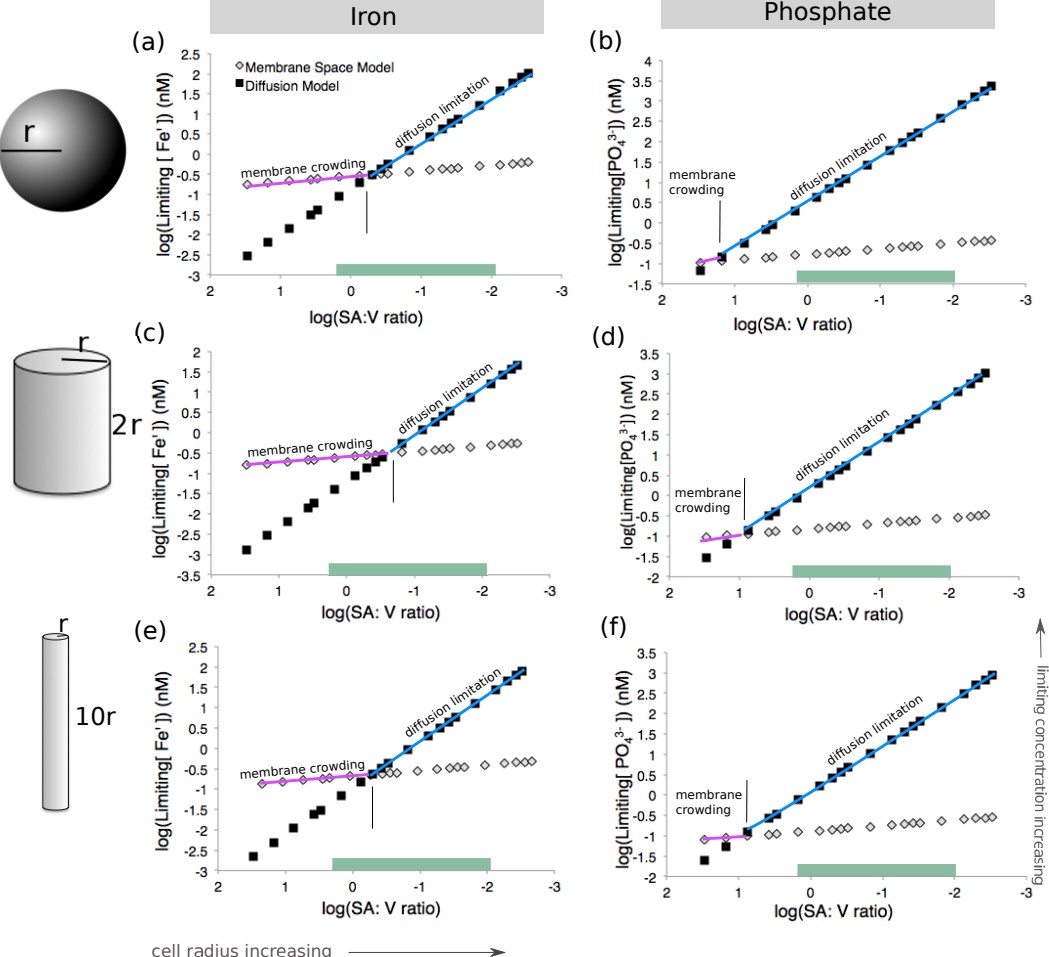

Figure 7. Model calculations for membrane space and diffusion based nutrient limitation reveal that membrane crowding could drive *Trichodesmium* to iron or phosphate stress, particularly when cells are small. Two cell morphologies (sphere and cylinder) were modeled for both iron and phosphate limitation. Calculations are detailed in Table S5. As the cell radius increases and the surface area: volume quotient decreases, the limiting concentration increases. This is concurrent with the current understanding that the low surface area: volume quotient of large cells leads to limitation. Green bars represent common SA: V ratio quotients for *T. Theibautii*.(Hynes et al., 2012) (A-B). Membrane crowding (purple) occurs if the limiting nutrient concentration is greater than in the diffusion limitation model (blue). Membrane crowding is more significant for cylindrical cells in particular (C-D); altering the length of the cylinder minimally affects the model (E-F).



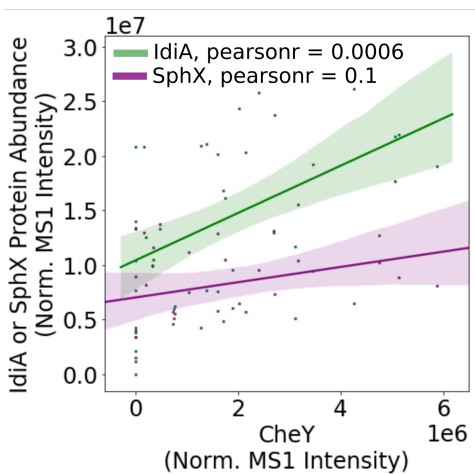

**Figure 8. CheY is positively correlated with the iron stress biomarker IdiA, but has a weaker association with phosphate stress biomarker SphX. This suggests that it might be involved in iron acquisition, for instance by helping colonies to move dust particles to the colony center. Pearson linear correlation coefficients (r values) are provided (p value for IdiA = 6e$^{-4}$, SphX = 0.1). Shaded bars indicate 95% confidence intervals.**

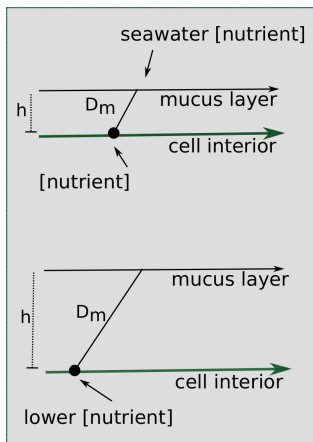

**Figure 9. Scheme for the effect of a mucus layer on nutrient diffusion. h = height of the mucus membrane, D$_m$ = diffusion coefficient of the mucus. Assuming some diffusion constant for the nutrient through the mucus and the same starting seawater nutrient concentration, a thicker layer of mucus surrounding a cell in a colony would result in a lower concentration of nutrient experienced at the cell surface.**





**Table 1. Quantification of the PstC ABC transporter and estimation of membrane space occupied**

| Station | [Pst] in fmol/ug total protein (replicate average) | standard deviation replicates (if available) | Pst molecules per cell assuming 30% w/w protein content* | % surface area occupied assuming 30% w/w^ | Pst molecules per cell assuming 55% w/w protein content** | % surface area occupied assuming 55% w/w^ |
|---|---|---|---|---|---|---|
| Tricolim_18 | 13.0 | 1.8 | 3.8E+05 | 3.6 | 7.0E+05 | 6.6 |
| Tricolim_15 | 11.2 | 3.4 | 3.3E+05 | 3.1 | 6.1E+05 | 5.7 |
| Tricolim_16 | 89.1 | 123.1 | 2.6E+06 | 24.5 | 4.8E+06 | 45.0 |
| JC150_3 | 38.7 | 63.3 | 1.1E+06 | 10.7 | 2.1E+06 | 19.5 |
| JC150_4 | 89.6 | 14.7 | 2.6E+06 | 24.7 | 4.9E+06 | 45.2 |
| JC150_5 | 74.2 | 36.4 | 2.2E+06 | 20.4 | 4.0E+06 | 37.5 |
| JC150_6 | 61.6 | 40.1 | 1.8E+06 | 17.0 | 3.3E+06 | 31.1 |
| JC150_7 | 165.7 | | 4.9E+06 | 45.6 | 9.0E+06 | 83.6 |
| JC150_1 | 106.1 | | 3.1E+06 | 29.2 | 5.7E+06 | 53.5 |
| **average** | | | | 19.9 | | 36.4 |
| **stdev** | | | | 13.4 | | 24.6 |

\* calculated using *Trichodesmium* cell volume of 3000um3 (Berman-Frank et al., 2001), cell volume to carbon conversion logC = 0.716log(V)-0.314 (Strathman, 1967), protein content of a cyanobacterium 30% w/w (Gonzalez Lopez et al., 2010), carbon to total protein conversion 0.53 g C/ g total protein (Rowenhoerst et al., 1991). \*\*calculated as in (\*) but with protein content of a cyanobacterium 55% w/w (Gonzalez Lopez et al., 2010). ^calculated using cross sectional area of an Ca ATPase of 0.0000167 um2 (Hudson and Morel 1992)

675