# Peer review of "Co-occurrence of Fe and P stress in natural populations of the"

_Biogeosciences, 2019_

## Referee Comment (RC1) · Anonymous Referee #1 · 18 Jan 2020

General overview

Held et al., present a metaproteomic analysis of Trichodesmium isolates mostly collected in the Tropical and Subtropical Atlantic and use targeted proteomics to quantify putative markers of P and Fe stress to assess correlations with Nitrogenase. Given that Trichodesmium is a dominant fixer of nitrogen in the oceans, mechanisms controlling the abundance the major nitrogen fixing enzyme within this cyanobacter is of great interest to ocean biogeochemists (and should be of interest to all). The study involves a survey of Trichodesmium populations and across multiple cruises and dates which supports a more generalizable perspective of Trichodesmium response to low Fe

or P. Rather than simply reporting data and immediate findings regarding correlations between proteins between P and Fe levels, the authors make a considerate attempt to extend hypotheses into the biophysical realm by setting up hypothetical scenarios of protein space competition that may lead to reduction in N fixation via membrane overcrowding. Despite lack of experimental evidence and sometime speculative, the authors cite precedent in many cases and have written a thought-provoking manuscript that should lead to testing of alternative hypotheses with regards to how Trichodesmium counters P and Fe limitations at the membrane in addition to regulatory gene expression.

Major comments

Abstract line 19, 20 the authors make reference to a '...specific physiological state under nutrient stress'. Given the fact that everything will have a specific physiological state under stress or non-stress conditions, this statement does not carry much impact. Perhaps the authors could be more specific. Did they intend to say that there is a generalized stress response regardless of stressors and that nitrogenase appears to comprise this stress response?

Intro Line 47 – The use of the term 'established' when describing biomarkers is a bit subjective and loses impact when not followed by multiple citations, which in turn, lends credence to the idea that the protein markers are routinely accepted and utilized by the scientific community. Its reasonable for the authors to state that these proteins were utilized as biomarkers in this study based upon scientific precedent from expression studies, but the discriminatory power of these proteins to classify has not been thoroughly validated. The rationale provided in line 47-59 is sufficient for inclusion in the study. The word putative or candidate in lieu of 'established' seems a better fit.

Line 151 -157. Point of clarification. The authors state that standard MS2 peak area was linear between 1 amol and 20 fmol per uL. Further that samples were spiked with standard to 10 fmol per uL and 10uL injected. The assumption is that 10uls was also

injected for linearity testing of the standards. Please confirm.

Methods: There is no mention of transition ions in the PRM section of the methods. For the sake of reproducibility, the authors should make reference to which transition ions were utilized for quantification (can be supplemental). Further the Trichodesmium genome and version should be referenced that was used for searching.

Line 196 "This clustering indicated direct regulatory links between C and N fixation.." perhaps use suggests in lieu of indicates due to probabilistic nature of the association of protein covariance. Similar strong language should be avoided without follow-up direct experimentation which is beyond the scope of the study.

Line 228 – 229 "This observation contrasts with the current paradigm that Trichodesmium down regulates nitrogen fixation when it is Fe or P stressed..." This statement needs to cite the current published paradigm.

Line 267 – 278 when considering membrane protein space, the authors make a considerate attempt. Space and physics are often ignored; however, because the authors measured whole cell protein abundance and did not attempt to isolate the plasma membrane fraction, the % occupancy estimates are bound to be overestimates. A statement providing limitations of the estimate are needed here. Limitation of space and crowding on a membrane makes sense. Assigning the protein number to the membrane alone is not accurate.

Line 321 "Thus, we conclude that in certain scenarios, lack of membrane space could indeed limit Fe and perhaps P acquisition by Trichodesmium." There is no disagreement with the rationale and calculations that led to this statement, but the statement is hypothetical in nature and the term 'hypothetical' should be included. The authors do a nice job in addressing model limitations in the next paragraph and go on to describe an artificial scenario where space limitation could produce further nutrient uptake limitation as additional proteins are made and transported to the membrane. One has to assume uptake activity does not change or is influenced by intracellular events or

membrane compositional changes that lead to conformational changes; however, the idea that a generalized stress response to Fe or P could lead to a negative effect on uptake and more limitation due to space limits is fun to ponder.

Line 369 – 385, Conclusions. The conclusion section is written as a perspective which is fine given that the conclusions linked specifically to the data are stated within the results and discussion. In line 372, the use of the word 'norm' is understandable given the common phrase 'the norm rather than the exception', but this might be contentious because normal is being assigned to all Trichodesmium based on 16 sampling sites mainly focused in the Atlantic. Fe and P stress may be more common that previously accepted or realized. If the authors feel strongly about this phrasing and believe the audience will be receptive and not over-interpret, then it is fine. Otherwise, perhaps it can be a bit more tempered. The same phrase was used in the Abstract, but due to space limitations the fact that authors were inferring norm based on their samples seems reasonable.

Conclusion Line 378-385, membrane space limitation is likely to be confronted by all cells, not only Trichodesmium, but the idea is understood. The authors make a case for including co-stress based on protein observations and if nutrient stress is truly occurring (which can be difficult to define to everyone's liking and measure in situ; hence use of protein markers) then including these parameters in biological models makes sense.

Figure 3. This is a very informative and nice figure. If possible it would be great to see Figure S2 incorporated for ease of reading and comparison.

Figure 9 is unnecessary, but certainly would be welcomed by visual learners if space is not an issue. Otherwise the concepts are described in the discussion.

Minor comments

Line 63-64"..suggesting nutrient stress was driven not only by biogeochemical gradients but also by Trichodesmium's inherent physiology". The term 'inherent physiology' is very broad and does not add substance to the sentence. Trichodesmium is responding to stress in the study and saying something like "…..but also by Trichodesmium's response to stress" puts the sentence in a category that doesn't contain every biochemical reaction in the organism.

Line 75. Table S1 does not correspond to the supplemental table indicated. Please correct the designation.

Line 100 "…...vacuum centrifugation to 1 100 $\mu$g $\mu$L-1 concentration." Assume this was estimated based on starting concentration of protein and not actually measured?

Line 107 "C18 columns packed in house." Please add column size, diameter, c18 particle size and supplier.

Line 110 A little more detail regarding the parameters would be useful. Based on the search parameters the instrument was likely operated in orbitrap/ion trap mode with HCD? This would be of interest to include and assume more details are in located in Pride.

Line 120 should include the term "local FDR" if local FDR was used.

Line 179-180 is repeated from methods section. Can be deleted.

Line 185 ribosomal and phycobilisomal

Line 237 – 257 This is quite fascinating although not the focus of the study. urtA substrate specificity is poorly defined outside of urea. Curious if the authors also found urease protein elevation in stressed samples.

Figure S2 "Note that the phosphate concentrations from the Tricolim cruise were not measured at the…" Do the authors mean to say Tricolim_13?

Supplemental figures. Please read through legends for spelling errors.

---

## Referee Comment (RC2) · Anonymous Referee #2 · 14 Feb 2020

General

The authors present a metaproteomic study of field-collected Trichodesmium colonies, focused on phosphate and iron stress markers, and complement that study with a membrane crowding model, which I think is a nice approach to try and understand the observed co-limitation patterns for iron and phosphate. The study is comprehensive in that samples from multiple cruises and years are used; with all but one station (HOT) located in the Atlantic Ocean. Increasing the knowledge of nutrient limitation in natural Trichodesmium populations is certainly of interest, given that it seems to be connected to aggregation of Trichodesmium in some way, either directly or through a general

stress response. While the study as such is valuable and should be published, I have a few major remarks that I think should be addressed before it is ready.

Major Remarks

1. The whole conclusion of co-occurring phosphate and iron stress relies on the assumption that protein abundances of IdiA and SphX are good proxies for iron or phosphate limitation, respectively. The authors do cite the relevant literature that showed upregulation of the respective markers under the corresponding nutrient stress. What I am missing is information on which fold-changes in protein abundance were measured in the cited studies under the respective nutrient limiting conditions. For example, what are the base levels of IdiA and SphX protein in the cell? If there is three times more SphX than IdiA, such as in Fig 3 for some of the Tricolim samples, does that really indicate co-limitation, or does that just reflect the base level of IdiA? For example, Snow et al, 2015 (Fig. 4) only report a two-fold change for IdiA from $\sim$100 fmol/ug to $\sim$200 fmol/ug under iron stress. I suggest presenting the evidence for IdiA and SphX being markers for the respective stresses clearly in a table, including the type of experiment (culture or field), absolute quantifications if stated, and fold-changes measured. I also suggest then being a little more careful in wording throughout the paper, and differentiating better how the results from different stations could be interpreted.

2. Figure 3, and the corresponding Figure S2 are nice and the basis for some important claims being made in part 3.2. of this manuscript. However, these claims should be supported with the necessary statistics, and it would help if Fig S2 was not in the Supplement, but presented together. For example, in line 210 ff, the authors claim that a) "Biomarkers for iron (IdiA) and phosphate (SphX) stress were highly abundant and positively associated with surface Fe or P concentrations" and b) "IdiA varied up to 8 fold, and increased moving West to East across the JC150 transect, consistent with an observed decrease in dFe concentrations" . For a) I think the authors mean "negatively", not "positively", correlated. And while I believe this correlation for SphX, it is not obvious for IdiA. For b) I cannot see increasing protein abundance from west to

east at all. Please prove this statistically before claiming it.

3. Given that only 1 sampling station is NOT in the Atlantic, please remove all claims that generalize the findings, e.g. "co-stress is the norm rather than the exception" (l. 18) → add "in the Atlantic", if wanting to keep this. Or in line 60f: "simultaneously Fe and P stressed throughout the worlds oceans" – this statement cannot be made with just one station outside the Atlantic.

Specific comments

Abstract

The abstract is missing some specificity. line 19: nitrogenase was most abundant – compared to what? Please rephrase: more abundant than under . . .

line 22: is confronted by the biophysical limits – when? Under which conditions is it confronted by this?

line 24f: be more specific. The last sentence is true for any microbe.

Introduction

Line 36: colloquialism

Line 58: add . . .Pho box, a regulatory DNA sequence, which is necessary. . .

Line 65f: Fe and P stress were positively associated – only as co-stress? If yes, say so. Also say how Fe, P, and N statuses are closely linked.

Methods

Line 119: what does that mean? Which precursors, of what? Was every protein normalized to the top 3 precursor intensities? Make this clear also to a reader who is not familiar with the specifics of proteomics analysis.

Line 120: How is the FDR defined? What does "0.1% peptide" and "1.2% protein" mean?

Line 128: Which peptides were selected?

Results and Discussion

Line 178: change "most" to "all but one"

Line 232: please rephrase. What exactly is common in marine bacteria. For sure, all bacteria have regulatory networks.

Line 255ff: skip this justification sentence

Section 3.4. Throughout this section, I think the use of the term ligand is not the norm. For ABC transporters, the word "ligand" is typically used for whatever binds to and is transported by the transporter. The part of the transporter binding the substrate is usually called "ligand-binding protein".

Line 260: change to "...required for both iron and phosphate uptake"

Line 305ff: rewrite sentence. Hard to understand.

Line 316ff on cylinders: Shouldn't the Trichodesmium filament, instead of a single Trichodesmium cell, be considered for these models? The effective cell surface of a Trichodesmium cell is reduced by its contact to the neighboring cells.

Line 361: Reference missing for mucus production being a "hallmark of Trichodesmium colony formation"

Line 362f: If mucus acts as a diffusive barrier, it also does the opposite of "protecting them [the cells] from oxygen", namely preventing O2 to diffuse out of the cells during photosynthesis, which was also shown in Eichner et al, 2019.

Line 384: Which specific regulatory systems should be characterized? What do you mean by chemical phases?

Figures

Figure 1: Please use the same numbers on the figure and the legend, or at least also

add the figure numbering top the legend.

Figure 2: Please increase the font size on the legend, and add a legend name like "# of times a protein appeared in the same cluster" – consider changing the legend to a percentage. Please also say in the caption what the color legend shows.

Figure 3: Please state in the caption how the protein abundance values were normalized.

Figure 4: Please adjust font sized throughout panels. How were the dashed lines in c and d defined? Based on what do they denote Fe- or P-stress? And why are they different in c and d?

Figure 5: Does not necessarily need to be a figure if wanting to save space.

---

## Author Comment (AC1) · 6 Mar 2020

We thank the reviewer for their thoughtful comments on our manuscript, particularly attention paid to the proteomics methodology. Responses to individual comments are provided below, including our proposed updates to the text.

Major comments

Abstract line 19, 20 the authors make reference to a '...specific physiological state under nutrient stress'. Given the fact that everything will have a specific physiological state under stress or non-stress conditions, this statement does not carry much impact.

Perhaps the authors could be more specific. Did they intend to say that there is a generalized stress response regardless of stressors and that nitrogenase appears to comprise this stress response?

We agree with the reviewer on this important clarification. The intention was to indicate a specific physiological state under co-stress, distinct from that of an organism stressed for just one nutrient. We have added a phrase to clarify this sentence:

Abstract. Trichodesmium is a globally important marine microbe that provides fixed nitrogen (N) to otherwise N limited ecosystems. In nature, nitrogen fixation is likely regulated by iron or phosphate availability, but the extent and interaction of these controls are unclear. From metaproteomics analyses using established protein biomarkers for iron and phosphate stress, we found that co-stress is the norm rather than the exception for field Trichodesmium colonies. Counter-intuitively, the nitrogenase enzyme was more abundant under co-stress than under single nutrient stress, consistent with the idea that Trichodesmium has a specific physiological state under nutrient co-stress, as opposed to single nutrient stress. Organic nitrogen uptake was observed to occur simultaneously with nitrogen fixation. Quantification of the phosphate ABC transporter PstA combined with a cellular model of nutrient uptake suggested that Trichodesmium is generally confronted by the biophysical limits of membrane space and diffusion rates for iron and phosphate acquisition in the field. Colony formation may benefit nutrient acquisition from particulate and organic nutrient sources, alleviating these pressures. The results highlight that to predict the behavior of Trichodesmium, both Fe and P stress must be evaluated and understood simultaneously.

Intro Line 47 – The use of the term 'established' when describing biomarkers is a bit subjective and loses impact when not followed by multiple citations, which in turn, lends credence to the idea that the protein markers are routinely accepted and utilized by the scientific community. Its reasonable for the authors to state that these proteins were utilized as biomarkers in this study based upon scientific precedent from expression studies, but the discriminatory power of these proteins to classify has not been thoroughly validated. The rationale provided in line 47-59 is sufficient for inclusion in the study. The word putative or candidate in lieu of 'established' seems a better fit.

We thank the reviewer for highlighting that protein biomarkers are not yet routinely utilized in the community. Change accepted:

There are several established protein biomarkers for Fe and P stress in Trichodesmium, all of which are periplasmic binding proteins involved in nutrient acquisition.

Line 151 -157. Point of clarification. The authors state that standard MS2 peak area was linear between 1 amol and 20 fmol per uL. Further that samples were spiked with standard to 10 fmol per uL and 10uL injected. The assumption is that 10uls was also injected for linearity testing of the standards. Please confirm.

Indeed we confirm that 10uL injections were used for linearity testing of the standard peptides. We now clarify this in the text:

The standard mixture was calibrated to establish the exact concentration of the peptides. A known amount (10 fmol $\mu$L-1) of the commercially available Pierce standard peptide mixture (Catalog number 88320) and an apomyoglobin digest was spiked into the standard. The ratio of Pierce (isotopically labelled according to JPT standards) or apomyoglobin (light) to heavy standard peptide MS2 peak area was calculated and used to establish the final concentration of the standard peptide mixture (Fu et al., 2016; Milo, 2013). Multiple peptides were used for this calibration and the standard deviation among them was approximately 10%. Finally, the linearity of the peptide standard was tested by generating a dilution curveand ensuring that the concentration of each peptide versus MS2 peak area was linear between 0.001 and 20 fmol $\mu$L-1 concentration, using 10uL injections consistent with experimental injection volumes.

Methods:

There is no mention of transition ions in the PRM section of the methods. For the sake of reproducibility, the authors should make reference to which transition ions were

utilized for quantification (can be supplemental). Further the Trichodesmium genome and version should be referenced that was used for searching.

An additional supplemental file (Table S6) providing the transition ions used for PstA quantification will now be provided. The Trichodesmium genome used for the initial (DDA) search is referenced in Section 2.4. In PRM mode specific masses were targeted, and the list of targeted ions will now be provided (Table S7). In the process of compiling these tables we noticed a mistake in our prior manuscript – the peptide quantified belongs to PstA, not PstC. Both PstA and PstC are components of the same phosphate transport permease protein, so results and discussion are not affected however it is important to make this distinction, and this naming has been updated in the text.

Line 196 "This clustering indicated direct regulatory links between C and N fixation.." perhaps use suggests in lieu of indicates due to probabilistic nature of the association of protein covariance. Similar strong language should be avoided without follow-up direct experimentation which is beyond the scope of the study.

Agreed and updated, particularly in this section (3.1) which relies on multivariate statistics: A self-organizing map analysis identified groups of proteins with similar profiles, i.e. proteins whose abundances changed cohesively, suggestive of proteins that may be regulated similarly (Reddy et al., 2016). This revealed the central importance of nitrogen fixation to Trichodesmium. The nitrogenase proteins were among the most abundant in the proteome and were located in clusters 1 and 2 (Figure 2 and Table S3). Also in these clusters were nitrogen metabolism proteins including glutamine synthetase, glutamine hydrolyzing guanosine monophosphate (GMP) synthase and glutamate racemase. This is consistent with previous reports finding that N assimilation is synchronized with nitrogen fixation (Carpenter et al., 1992).

Nitrogen fixation was closely linked to carbon fixation. Many photosystem proteins clustered with the nitrogenase proteins, including phycobilisome proteins, photosystem

proteins, and the citric acid cycle protein 2-oxoglutarate dehydrogenase. This clustering indicated direct regulatory links between C and N fixation. The nitrogen regulators P-II and NtcA were also present in this cluster and may mediate this association. In non-nitrogen fixing cyanobacteria, high abundance of the nitrogen regulators NtcA and P-II is suggestive of nitrogen stress (Flores and Herrero, 2005; Saito et al., 2014). In diazotrophs, the role of these regulators is unclear because they do not respond to nitrogen compounds such as ammonia as they do in other cyanobacteria (Forchhammer and De Marsac, 1994). Here, clustering of NtcA and P-II with C and N fixation proteins suggests that they play a role in balancing these processes in field populations, though the details of this role has yet to be elucidated.

Line 228 – 229 "This observation contrasts with the current paradigm that Trichodesmium down regulates nitrogen fixation when it is Fe or P stressed. . ." This state- ment needs to cite the current published paradigm. Change accepted: This observation contrasts with the current paradigm that Trichodesmium down regulates nitrogen fixation when it is Fe or P stressed (Frischkorn et al., 2018, Ruoco, et al., 2018, Bergmann et al., 2012, Shi et al., 2007).

Line 267 – 278 when considering membrane protein space, the authors make a considerate attempt. Space and physics are often ignored; however, because the authors measured whole cell protein abundance and did not attempt to isolate the plasma membrane fraction, the % occupancy estimates are bound to be overestimates. A statement providing limitations of the estimate are needed here. Limitation of space and crowding on a membrane makes sense. Assigning the protein number to the membrane alone is not accurate.

This is an important point and a sentence has been added to the text to indicate this limitation:

"The calculation assumes that 100% of the PstC protein quantified is present in the plasma membrane, however it should be recognized that some fraction is likely present

in the cytoplasm, leading to the possibility of over-estimation of membrane space occupied."

Line 321 "Thus, we conclude that in certain scenarios, lack of membrane space could indeed limit Fe and perhaps P acquisition by Trichodesmium." There is no disagreement with the rationale and calculations that led to this statement, but the statement is hypothetical in nature and the term 'hypothetical' should be included. The authors do a nice job in addressing model limitations in the next paragraph and go on to describe an artificial scenario where space limitation could produce further nutrient uptake limitation as additional proteins are made and transported to the membrane. One has to assume uptake activity does not change or is influenced by intracellular events or membrane compositional changes that lead to conformational changes; however, the idea that a generalized stress response to Fe or P could lead to a negative effect on uptake and more limitation due to space limits is fun to ponder.

We thank the reviewer for their positive comment and interest in this section. We have updated Line 321 to highlight uncertainty in the statement:

Thus, we conclude that in certain scenarios, lack of membrane space could hypothetically limit Fe and perhaps P acquisition by Trichodesmium.

Line 369 – 385, Conclusions. The conclusion section is written as a perspective which is fine given that the conclusions linked specifically to the data are stated within the results and discussion. In line 372, the use of the word 'norm' is understandable given the common phrase 'the norm rather than the exception', but this might be contentious because normal is being assigned to all Trichodesmium based on 16 sampling sites mainly focused in the Atlantic. Fe and P stress may be more common that previously accepted or realized. If the authors feel strongly about this phrasing and believe the audience will be receptive and not over-interpret, then it is fine. Otherwise, perhaps it can be a bit more tempered. The same phrase was used in the Abstract, but due to space limitations the fact that authors were inferring norm based on their samples

seems reasonable.

We thank the reviewer for this comment. We have altered our wording in this section to avoid over-reach (similar changes will also be made elsewhere in the paper):

Trichodesmium's colonial lifestyle likely produces challenges for dissolved Fe and P acquisition, which must be compensated for by production of multiple nutrient transport systems, such as for particulate iron and organic phosphorous, at a considerable cost. While laboratory studies have largely focused on single nutrient stresses in free filaments, these metaproteomic observations and accompanying nutrient uptake model demonstrate that Fe and P co-stress may be the norm rather than the exception, particularly in the North Atlantic ocean. This means that the emphasis on single limiting nutrients in culture studies and biological models may not capture the complexities of Trichodesmium's physiology in situ. Thus, biogeochemical models should consider incorporating Fe and P co-stress conditions. Specifically, in this study and in others there is evidence that nitrogen fixation is optimal under co-limited or co-stressed conditions, implying that an input of either Fe or P could counter-intuitively decrease N2 driven new production (Garcia et al., 2015; Walworth et al., 2016).

Conclusion Line 378-385, membrane space limitation is likely to be confronted by all cells, not only Trichodesmium, but the idea is understood. The authors make a case for including co-stress based on protein observations and if nutrient stress is truly occurring (which can be difficult to define to everyone's liking and measure in situ; hence use of protein markers) then including these parameters in biological models makes sense.

We agree that membrane space is likely confronted by all cells, though few oceanographic studies have demonstrated or discussed this. In the interest of not over-reaching our argument we leave the reviewer's point out of our main text but hope that this work will stimulate future investigations.

Figure 3. This is a very informative and nice figure. If possible it would be great to see

Figure S2 incorporated for ease of reading and comparison.

Both reviewer #1 and reviewer #2 suggested this change, so we now include the Figure S2 plots in the main text as part of Figure 3. Figure S2 now provides scatter plots of IdiA relating dFe and SphX versus phosphate concentrations. Per these changes the text in Section 3.2 has been updated as below.

3.2 Trichodesmium is simultaneously iron and phosphate stressed throughout its habitat A surprising emergent observation from the Trichodesmium metaproteomes was the co-occurrence of the iron (IdiA) and phosphate (SphX) stress biomarkers across the samples. The ubiquitous and highly abundant presence of these proteins relative to total protein implied that co-stress may be the norm rather than the exception for Trichodesmium colonies in the field, particularly in the North Atlantic. Even though low-level basal expression of IdiA and SphX has been observed, it was clear that the colonies were devoting a large fraction of their cellular resources to Fe and P uptake, respectively (see Tables S8 and S9) (Webb et al., 2001, Webb et al., 2007, Chappell et al., 2010, Orchard et al., 2010, Snow et al., 2015, Walworth et al., 2016, Frischkorn et al., 2019). This, combined with the responsiveness of IdiA and SphX to nutrient availability in laboratory experiments on Trichodesmium filaments laboratory, indicated that co-stress was occurring.

Interestingly, biomarker abundance was not necessarily associated with nutrient concentrations in the surface ocean, suggesting that the colonies were experiencing stress despite variation in nutrient availability (Figure 3 C-D). SphX abundance varied up to 7.5 fold and may have been negatively associated with dissolved phosphate concentrations, though analytical differences across the field expeditions may have forced this relationship (Figure S2). Oceanographically, SphX was most abundant in the P-deplete, summer-stratified North Atlantic gyre (JC150 expedition) compared with winter waters near the Amazon river plume (Tricolim expedition) or at station ALOHA, where phosphate concentrations were greater (Hynes et al., 2009; Sañudo-Wilhelmy et al., 2001; Wu et al., 2000). IdiA varied up to 8 fold but there was no observable relationship with

dFe concentrations at the surface. Instead, IdiA may be responsive to other factors such as the varying iron requirements of the populations/species examined here. For instance, it should be highlighted that in this study only Trichodesmium colonies were examined, so factors such as colony size may affect iron availability and biomarker expression. Additionally, because the surface ocean iron inventory is low, transient inputs such as from the Sahara desert can dramatically impact iron availability on short time scales, and the time scale of these inputs relative to changes in biomarker abundance is not well understood (Kunde et al., 2019). Carefully calibrated datasets relating IdiA and SphX abundance to nutrient-limited growth rates of Trichodesmium in both the filamentous and colonial forms would facilitate further interpretation of this data.

Figure 9 is unnecessary, but certainly would be welcomed by visual learners if space is not an issue. Otherwise the concepts are described in the discussion.

As the first author is a visual learner and space is not an issue, she would prefer to leave this in, but welcomes editorial advice on the matter!

Minor comments Line 63-64"..suggesting nutrient stress was driven not only by biogeochemical gradi- ents but also by Trichodesmium's inherent physiology". The term 'inherent physiology' is very broad and does not add substance to the sentence. Trichodesmium is respond- ing to stress in the study and saying something like ". . ..but also by Trichodesmium's response to stress" puts the sentence in a category that doesn't contain every bio- chemical reaction in the organism.

Rephrased: "suggesting nutrient stress was driven not only by biogeochemical gradients but also by Trichodesmium's response to nutrient depletion"

Line 75. Table S1 does not correspond to the supplemental table indicated. Please correct the designation. Change accepted – should read Table S2

Line 100 ". . ...vacuum centrifugation to 1 100 $\mu$g $\mu$L-1 concentration." Assume this was estimated based on starting concentration of protein and not actually measured?

Yes, this is estimated from starting concentration and will now be noted in the text: The resulting peptide mixture was concentrated by vacuum centrifugation to 1 $\mu$g $\mu$L-1 concentration estimated from the starting protein concentration.

Line 107 "C18 columns packed in house." Please add column size, diameter, c18 particle size and supplier.

This section has been updated with more details per the reviewer:

2.3 Sample acquisition The global proteomes were analysed by online comprehensive active-modulation two-dimensional liquid chromatography (LC x LC-MS) using high and low pH reverse phase chromatography with inline PLRP-S (200$\mu$m x 150mm, 3$\mu$m bead size, 300A pore size, NanoLCMS Solutions) and C18 columns packed in house (100 ïA■m x 150 mm, 3 $\mu$m particle size, 120 Å pore size, C18 Reprosil-Gold, Dr. Maisch GmbH packed in a New Objective PicoFrit column). The first dimension utilized an 8 hour pH = 10 gradient (10mM ammonium formate and 10mM ammonium formate in in 90% acetonitrile), and was trapped every 30min on alternating dual traps, then eluted at 500nL/min onto the C18 column with a 30 min gradient (0.1% formic acid and 0.1% formic acid in 99.9% acetonitrile). 10 $\mu$g of protein was injected per run directly onto the first column using a Thermo Dionex Ultimate3000 RSLCnano system (Waltham, MA), and an additional RSLCnano pump was used for the second dimension gradient. The samples were then analyzed on a Thermo Orbitrap Fusion mass spectrometer with a Thermo Flex ion source (Waltham, MA). MS1 scans were monitored between 380-1580 m/z, with a 1.6 m/z MS2 isolation window (CID mode), 50 millisecond maximum injection time and 5 second dynamic exclusion time.

Line 110 A little more detail regarding the parameters would be useful. Based on the search parameters the instrument was likely operated in orbitrap/ion trap mode with HCD? This would be of interest to include and assume more details are in located in Pride.

Details are located in Pride but the reviewer is right that more should be included in the

main text. Added sentence: "MS1 scans were monitored between 380-1580 m/z, with a 1.6 m/z MS2 isolation window (CID mode), 50 millisecond maximum injection time and 5 second dynamic exclusion time."

Line 120 should include the term "local FDR" if local FDR was used.

These are global FDRs (will now be noted).

Line 179-180 is repeated from methods section. Can be deleted.

Change accepted

Line 185 ribosomal and phycobilisomal

Change accepted

Line 237 – 257 This is quite fascinating although not the focus of the study. urtA substrate specificity is poorly defined outside of urea. Curious if the authors also found urease protein elevation in stressed samples.

Unfortunately urease protein abundances were very patchy (and were generally in low abundance) so we were unable to draw any specific conclusions. We did make an effort to note that increased abundance of the urea ABC transporter indicates general use of organic N sources including urea but also possibly TMA or other compounds.

Figure S2 "Note that the phosphate concentrations from the Tricolim cruise were not measured at the. . ." Do the authors mean to say Tricolim_13?

Per the reviewers we now include the data Figure S2 in the main text as part of Figure 3, and have updated the caption accordingly.

Supplemental figures. Please read through legends for spelling errors.

Change accepted.

Please also note the supplement to this comment:

https://www.biogeosciences-discuss.net/bg-2019-493/bg-2019-493-AC1-supplement.pdf

**BGD**

---

## Author Comment (AC2) · 6 Mar 2020

We thank the reviewer for their thoughtful comments on our manuscript and discuss changes below. We also provide a supplemental file with new tables/figures to be included in the manuscript, as well as the following responses formatted for ease of reading.

Major Remarks

1. The whole conclusion of co-occurring phosphate and iron stress relies on the assumption that protein abundances of IdiA and SphX are good proxies for iron or phos-

phate limitation, respectively. The authors do cite the relevant literature that showed upregulation of the respective markers under the corresponding nutrient stress. What I am missing is information on which fold-changes in protein abundance were measured in the cited studies under the respective nutrient limiting conditions. For example, what are the base levels of IdiA and SphX protein in the cell? If there is three times more SphX than IdiA, such as in Fig 3 for some of the Tricolim samples, does that really indicate co-limitation, or does that just reflect the base level of IdiA? For example, Snow et al, 2015 (Fig. 4) only report a two-fold change for IdiA from âĹij100 fmol/ug to âĹij200 fmol/ug under iron stress. I suggest presenting the evidence for IdiA and SphX being markers for the respective stresses clearly in a table, including the type of experiment (culture or field), absolute quantifications if stated, and fold-changes measured. I also suggest then being a little more careful in wording throughout the paper, and differentiating better how the results from different stations could be interpreted.

The reviewer asks us to address how our results compare with previous reports of IdiA and SphX abundance during nutrient limitation. Per the reviewer, we provide new supplementary tables (Table S8 and S9) containing fold-change values from the literature for IdiA and SphX as during nutrient limitation. The biomarkers increased in abundance during nutrient depletion, however the magnitude of the response varied. This is likely due to experimental differences such as the analytical methodology (Western blots versus LC-MS), nutrient concentrations or growth rate of the culture being examined. For instance, IdiA responded less strongly to iron stress in the Walworth et al., 2016 experiments compared to the Webb et al., 2001 and Snow et al., 2015 experiments, and we hypothesize this is because the Fe-depleted condition had 10 nM added iron as opposed to 0 nM iron in Webb et al. 2001 and Snow et al. 2015. We agree with the reviewer that applying a quantitative framework to this data would be valuable once the necessary data becomes available and now note this in the text.

Based on the consistent responsiveness of IdiA and SphX to nutrient limitation in the laboratory, we concluded that the high relative abundance of these biomarkers was indicative of nutrient stress. The reasoning is that the cells were clearly devoting a large fraction of their proteome to Fe and P uptake, likely at the expense of other nutrient uptake systems such as for organic nitrogen as is discussed later in the text. The reviewer's point that there may be basal expression of IdiA and SphX in replete cells is well taken. Because we report relative abundance data, we cannot directly compare our results to those of other researchers who took different quantitative approaches. However, the possibility of basal expression should be clarified in the text. Carefully calibrated datasets relating IdiA and SphX protein abundance to nutrient limited growth rates, while outside the scope of this paper, would be valuable in facilitating interpretation of this data. Based on the reviewer's comments we have updated Section 3.2 below to clarify the assumptions and caveats in our interpretation, and to highlight the need for calibrated biomarker studies.

3.2 Trichodesmium is simultaneously iron and phosphate stressed throughout its habitat

A surprising emergent observation from the Trichodesmium metaproteomes was the co-occurrence of the iron (IdiA) and phosphate (SphX) stress biomarkers across the samples. The ubiquitous and highly abundant presence of these proteins relative to total protein implied that co-stress may be the norm rather than the exception for Trichodesmium colonies in the field, particularly in the North Atlantic. Even though low-level basal expression of IdiA and SphX has been observed, it was clear that the colonies were devoting a large fraction of their cellular resources to Fe and P uptake, respectively (see Tables S8 and S9) (Webb et al., 2001, Webb et al., 2007, Chappell et al., 2010, Orchard et al., 2010, Snow et al., 2015, Walworth et al., 2016, Frischkorn et al., 2019). This, combined with the responsiveness of IdiA and SphX to nutrient availability in Trichodesmium filaments in the laboratory, indicated that co-stress was occurring.

Interestingly, biomarker abundance was not necessarily associated with nutrient concentrations in the surface ocean, suggesting that the colonies were experiencing stress

despite variation in nutrient availability (Figure 3 C-D). SphX abundance varied up to 7.5 fold and were negatively associated with dissolved phosphate concentrations, though analytical differences across the field expeditions may have forced this relationship (Figure S2). Oceanographically, SphX was most abundant in the P-deplete, summer-stratified North Atlantic gyre (JC150 expedition) compared with winter waters near the Amazon river plume (Tricolim expedition) or at station ALOHA, where phosphate concentrations were greater (Hynes et al., 2009; Sañudo-Wilhelmy et al., 2001; Wu et al., 2000). IdiA varied up to 8 fold but there was no observable relationship with dFe concentrations at the surface. Instead, IdiA may be responsive to other factors such as the varying iron requirements of the populations/species examined. It should be highlighted that in this study only Trichodesmium colonies were considered, so factors such as colony size may affect iron availability and biomarker expression. Additionally, because the surface ocean iron inventory is low, transient inputs such as from the Sahara desert can dramatically impact iron availability on short time scales, and the time scale of these inputs relative to changes in biomarker abundance is not well understood (Kunde et al., 2019). Carefully calibrated datasets relating IdiA and SphX abundance to nutrient-limited growth rates of Trichodesmium in both the filamentous and colonial forms would facilitate further interpretation of this data.

2. Figure 3, and the corresponding Figure S2 are nice and the basis for some important claims being made in part 3.2. of this manuscript. However, these claims should be supported with the necessary statistics, and it would help if Fig S2 was not in the Supplement, but presented together. For example, in line 210 ff, the authors claim that a) "Biomarkers for iron (IdiA) and phosphate (SphX) stress were highly abundant and positively associated with surface Fe or P concentrations" and b) "IdiA varied up to 8 fold, and increased moving West to East across the JC150 transect, consistent with an observed decrease in dFe concentrations". For a) I think the authors mean "negatively", not "positively", correlated. And while I believe this correlation for SphX, it is not obvious for IdiA. For b) I cannot see increasing protein abundance from west Please prove this statistically before claiming it.

First, we thank the reviewer for correcting our mistake in line 210 – we did indeed mean to write "negatively." For clarity, we have moved the panels in Figure S2 alongside Figure 3 in the main text. Figure S2 now provides scatter plots of IdiA and SphX versus dFe and phosphate concentrations in the surface ocean (see updated figures section at the end of this document). There was a statistically observable relationship between SphX and dissolved phosphate, however the relationship may be forced by the different analytical approaches used on the JC150 versus Tricolim expeditions as is noted in the text below. By contrast there was no statistically observable relationship between IdiA and dFe, even though laboratory experiments clearly indicated that IdiA was a good biomarker of Fe stress in Trichodesmium. There are many factors that could influence this association, particularly iron speciation, changes in iron quotas, and factors such as colony size, which are not controlled in the field. We thank the reviewer for calling this point to our attention as it provides an opportunity to discuss these points in the updated Section 3.2 (see Major Remarks #1).

3. Given that only 1 sampling station is NOT in the Atlantic, please remove all claims that generalize the findings, e.g. "co-stress is the norm rather than the exception" (l. 18) → add "in the Atlantic", if wanting to keep this. Or in line 60f: "simultaneously Fe and P stressed throughout the worlds oceans" – this statement cannot be made with just one station outside the Atlantic.

Change accepted.

Specific Comments

Abstract

The abstract is missing some specificity.

line 19: nitrogenase was most abundant – compared to what? Please rephrase: more abundant than under . . line 22: is confronted by the biophysical limits – when? Under which conditions is it confronted by this? line 24f: be more specific. The last sentence

is true for any microbe.

We agree with these points and have updated the text accordingly. On line 24f, we wished to highlight the importance of considering multiple nutrients for Trichodesmium specifically, given the historical emphasis on either Fe or P stress.

Abstract. Trichodesmium is a globally important marine microbe that provides fixed nitrogen (N) to otherwise N limited ecosystems. In nature, nitrogen fixation is likely regulated by iron or phosphate availability, but the extent and interaction of these controls are unclear. From metaproteomics analyses using established protein biomarkers for iron and phosphate stress, we found that co-stress is the norm rather than the exception for field Trichodesmium colonies. Counter-intuitively, the nitrogenase enzyme was more abundant under co-stress than under single nutrient stress, consistent with the idea that Trichodesmium has a specific physiological state under nutrient co-stress, as opposed to single nutrient stress. Organic nitrogen uptake was observed to occur simultaneously with nitrogen fixation. Quantification of the phosphate ABC transporter PstA combined with a cellular model of nutrient uptake suggested that Trichodesmium is generally confronted by the biophysical limits of membrane space and diffusion rates for iron and phosphate acquisition in the field. Colony formation may benefit nutrient acquisition from particulate and organic nutrient sources, alleviating these pressures. The results highlight that to predict the behavior of Trichodesmium, both Fe and P stress must be evaluated and understood simultaneously.

Introduction

Line 36: colloquialism

Change accepted.

Line 58: add . . .Pho box, a regulatory DNA sequence, which is necessary. . .

Change accepted

Line 65f: Fe and P stress were positively associated – only as co-stress? If yes, say

so.

They were associated both individually and under co-stressed.

Also say how Fe, P, and N statuses are closely linked.

We suggest these are linked via a currently unknown regulatory network; change accepted.

Methods

Line 119: what does that mean? Which precursors, of what? Was every protein normalized to the top 3 precursor intensities? Make this clear also to a reader who is not familiar with the specifics of proteomics analysis.

Relative abundance was measured by averaging the peptide precursor/MS1 intensities for the 3 most abundant peptides in the protein, then normalizing this value to the total precursor intensity. Text has been updated:

Raw spectra were searched with the Sequest algorithm using a custom-built genomic database (Eng, Fischer, Grossmann, and MacCoss, 2008). The genomic database consisted of a publically available Trichodesmium community metagenome available on the JGI IMG platform (IMG ID 2821474806), as well as the entire contents of the CyanoGEBA project genomes (Shih et al., 2013). Protein annotations were derived from the original metagenomes. SequestHT mass tolerances were set at +/- 10ppm (parent) and +/- 0.8 Dalton (fragment). Cysteine modification of +57.022 and methionine modification of +16 were included. Protein identifications were made with Peptide Prophet in Scaffold (Proteome Software) at the 95% protein and peptide identification levels. Relative abundance was measured by averaging the precursor intensity (area under the MS1 peak) of the top 3 most abundant peptides in each protein, then normalizing this value to total precursor ion intensity. Normalization and global false discovery rate (FDR) calculations, which were 0.1% at the peptide level and 1.2% at the protein level, were performed in Scaffold (Proteome Software). FDR

was calculated by Scaffold using the probabilistic method by summing the assigned protein or peptide probabilities and dividing by the maximum probability (100%) for each. The mass spectrometry proteomics data have been deposited to the ProteomeXchange Consortium via the PRIDE partner repository with the dataset identifier PXD016225 and 10.6019/PXD016225 (Perez-Riverol et al., 2019). Statistical tests of relationships between proteins were conducted with the scipy stats package (https://docs.scipy.org/doc/scipy/reference/stats.html) using linear Pearson tests when the relationship appeared to be linear and a Spearman rank order test when this was not the case.

Line 120: How is the FDR defined? What does "0.1% peptide" and "1.2% protein" mean?

Text updated (see above paragraph):

FDR was calculated by Scaffold using the probabilistic method by summing the assigned protein probabilities and dividing by the maximum probability (100%) for each. Different FDRs can be assigned for peptides versus proteins depending on which probabilities are used for the calculation.

Line 128: Which peptides were selected? We have added to the supplemental Tables S6 and S7 which describe the peptides selected for quantitation

Results and Discussion

Line 178: change "most" to "all but one"

Change accepted

Line 232: please rephrase. What exactly is common in marine bacteria. For sure, all bacteria have regulatory networks.

Change accepted. A recent review of regulatory genes found that regulatory networks may be particularly more abundant in marine organisms (Held et al., 2019).

This indicates that the cell's N, P and Fe statuses are linked, perhaps involving a regulatory network which are particularly common in marine bacteria (Figure 5) (Held et al., 2019).

Line 255ff: skip this justification sentence

With respect we prefer to leave this sentence in because exploratory metaproteomics is not yet widely used as an analytical tool in oceanography. We hope this study and others like it will encourage its adoption. We do welcome editorial advice or further discussion on this point.

Section 3.4. Throughout this section, I think the use of the term ligand is not the norm. For ABC transporters, the word "ligand" is typically used for whatever binds to and is transported by the transporter. The part of the transporter binding the substrate is usually called "ligand-binding protein".

We agree with the reviewer that "ligand" is often used to describe a chemical compound, for instance a siderophore, which can be transported by an ABC transporter. However specifically in the uptake kinetics literature "ligand" is used to describe the ABC transporter itself (i.e. the protein that binds the nutrient). To avoid confusion we have updated the text to use the word "transporter" or "protein" instead.

Line 260: change to ". . .required for both iron and phosphate uptake"

Change accepted

Line 305ff: rewrite sentence. Hard to understand.

Updated, and hopefully clearer now!

For a given surface area: volume quotient, we define nutrient limitation to be caused by either membrane crowding or diffusion limitation depending on which model calculated a higher minimum nutrient concentration.

Line 316ff on cylinders: Shouldn't the Trichodesmium filament, instead of a single Trichodesmium cell, be considered for these models? The effective cell surface of a Trichodesmium cell is reduced by its contact to the neighboring cells.

The reviewer is correct that membrane limitation would be exacerbated for cells living in filaments, as the surface area exposed to the surrounding environment would be reduced. We considered modeling filamentous cells but decided to consider only single cells for clarity since this is the most conservative scenario (i.e. the one in which Trichodesmium would theoretically have the most exposure to the environment and be the least limited). We have added a sentence to the discussion of the model highlighting the focus of the model (single cells) but mentioning that filamentous cells would have lower surface area.

While this model may be directly applicable to some N2-fixing cyanobacteria such as Groups B and C, which have roughly spherical cells, Trichodesmium cells are not spheres but rather roughly cylindrical (Hynes et al., 2012). Thus, we repeated the model calculations for cylinders with varying radii (r) and heights (2r or 10r) based on previous estimates of Trichodesmium cell sizes (Bergman et al., 2013; Hynes et al., 2012). Cylinders have lower surface area: volume quotient than spheres of similar sizes. In addition, the rate constant (kD) for diffusion, which is a function of cell geometry, is greater. This increases the slope of the diffusion limitation line such that membrane crowding is important across a greater range of cell sizes (Figure 7c-d). Trichodesmium cell sizes vary in nature, for instance the cylinder height can be elongated, improving the surface area: volume quotient. However, the impact of cell elongation to radius r and height 10r on both diffusion limitation and membrane crowding is subtle (Figure 7e-f). Furthermore, though not explicitly considered here, cylindrical cells living in filaments would have reduced surface area available for nutrient uptake. Thus, we conclude that in certain scenarios, lack of membrane space could hypothetically limit Fe and perhaps P acquisition by Trichodesmium.

Line 361: Reference missing for mucus production being a "hallmark of Trichodesmium colony formation"

Citation added (Eichner et al., 2019)

Line 362f: If mucus acts as a diffusive barrier, it also does the opposite of "protecting them [the cells] from oxygen", namely preventing O2 to diffuse out of the cells during photosynthesis, which was also shown in Eichner et al, 2019.

Note added to this effect:

A key hallmark of Trichodesmium colony formation is production of mucus, which can capture particulate matter and concentrate it within the colony (Eichner et al., 2019). In addition to particle entrainment, the mucus layer can benefit cells by protecting them from oxygen and/or concentrating oxygen during photosynthesis, facilitating epibiont associations, regulating buoyancy, defending against grazers and helping to "stick" trichomes together (Eichner et al., 2019; Lee et al., 2017; Sheridan, 2002). However, these benefits come at a cost because the mucus layer hinders diffusion to the cell surface (Figure 9), reducing contact with the surrounding seawater. Despite this, the benefits of colony formation seem to outweigh the costs, since Trichodesmium forms colonies in the field, particularly under stress (Bergman et al., 2013; Capone et al., 1997; Hynes et al., 2012).

Line 384: Which specific regulatory systems should be characterized? What do you mean by chemical phases?

We don't know yet which regulatory systems should be examined! For a review of marine regulatory systems and their often unknown functions see Held et al. 2019. These results suggest that one or more regulatory networks may control Fe, P, and N status in tandem with one another in Trichodesmium cells, and we hope this work will stimulate future research on this topic. By chemical phases we meant dissolved versus particulate nutrient sources, since Trichodesmium is known to use both – we've clarified this now:

Future studies should aim to characterize the specific regulatory systems, chemical

species and phases (i.e. dissolved versus particulate nutrient sources), and symbiotic interactions that underlie Trichodesmium's unique behavior and lifestyle.

Figures

Figure 1: Please use the same numbers on the figure and the legend, or at least also add the figure numbering top the legend.

Figure 1 has been updated (see supplemental file).

Figure 2: Please increase the font size on the legend, and add a legend name like "# of times a protein appeared in the same cluster" – consider changing the legend to a percentage. Please also say in the caption what the color legend shows.

Figure 2 and its caption have been updated

Figure 3: Please state in the caption how the protein abundance values were normalized.

Figure 3 caption has been updated (see supplemental file)

Figure 4: Please adjust font sized throughout panels. How were the dashed lines in c and d defined? Based on what do they denote Fe- or P-stress? And why are they different in c and d?

Figure 4 font sizes have been updated. The dashed lines in (C) and (D) were drawn by hand to help the reader to visually understand the intersection of Fe and P stress. However, given the important discussion about IdiA and SphX abundances raised by the reviewer, we can see how this might be misleading and have removed the dotted lines.

Figure 5: Does not necessarily need to be a figure if wanting to save space.

We welcome editorial advice on this but would advocate for including the figure as we think it illustrates the discussion in section 3.3, particularly for visual learners.

[Figure]

Please also note the supplement to this comment:
https://www.biogeosciences-discuss.net/bg-2019-493/bg-2019-493-AC2-supplement.pdf
* * *
[Figure]

**Supplement:**

**Response to Reviewers**

**Reviewer # 2**

**Updated figures and tables to be included**

[Figure]

Figure 1. Sampling locations. Red/pink colors indicate JC150 stations; blue colors indicate Tricolim stations, dark grey indicates the Bermuda Atlantic Time Series (BATS) and light grey indicates Hawaii Ocean Time Series (HOT). Most samples exist in duplicate or triplicate; see Table S2 for detailed information.

[Figure]

Figure 2. Heatmap displaying results of self-organizing map analysis. Each protein was mapped to a self-organizing map grid, and the grids subsequently clustered by a k-means clustering algorithm. The process was repeated 10,000 times and the results displayed here as a heatmap with warm colors representing proteins that appear in the same cluster. The color bar indicates the percent of SOM experiments in which two proteins appear in the same cluster. Only the top 500 most abundant proteins are displayed. Dark yellow indicates proteins that appear in the same cluster 99.99% of the time. Clusters # 1 and 2 contain nitrogen fixation, carbon fixation, and nitrogen assimilation proteins as well as the regulatory systems NtcA and P-II. The cluster assignments for the proteins are available in Table S4.

[Figure]

Figure 3. (A) Relative abundance of iron stress protein IdiA (A) and phosphate stress protein SphX (B). IdiA and SphX were among the most abundant proteins in the entire dataset. Error bars are one standard deviation on the mean when multiple samples were available. Dashed lines represent average values across the dataset. Proteins abundances were normalized such that the total MS1 peak area across the entire proteome was the same for each sample. (C) and (D) concentrations of dFe and phosphate nutrients. (E) Relative abundance of IdiA (orange) and SphX (blue) overlaid on the sampling locations.

[Figure]

Figure 4. Nitrogenase abundance is highest at the intersection of high iron and phosphate stress. A) IdiA and B) SphX abundance is positively related to nitrogenase MoFe and Fe protein abundance (c = Spearman rank-order correlation coefficient, p = Spearman p-value). Effects of combined iron and phosphate stress biomarkers on nitrogenase abundance. Marker colors represent abundance of NifK (panel C) and NifH (panel D).

[Figure]

Figure S2. Scatter plot of A) SphX versus surface phosphate and B) IdiA versus surface dFe values. Note that analytical differences between JC150 (red dots) and Tricolim (blue dots) may be forcing the relationship with the surface phosphate values, though all values are above the limit of detection.

**New Supplementary Tables to be included**

**Table S6**. Transition ions used for PstA protein quantification (peptide ATDEALQIVPR)

| **Peptide ATDEALQIVPR** | y8 - 925.5465+ |
| --- | --- |
| | y7 - 796.5039+ |
| | b3 - 288.1190+ |
| | b4 - 417.1616+ |
| | b5 - 488.1987+ |
| | b6 - 601.2828+ |

**Table S7.** All peptides targeted in PRM mode (note only PstA data reported here)

| Description | Tery # | Metagenome ID | Peptide |
|---|---|---|---|
| IdiA periplasmic binding | Tery_3377 | TCCM_0877.00000020 | IFSEGNNEYPVVAGIPIATVLK |
| | | | HYDTDQALYDSFTQK |
| | | | FLEHLVSPEAQK |
| | | | ILYHDQNIYDPDIDPVEIR |
| phosphate ATP binding protein PstB | Tery_3540 | TCCM_0018.00000090 | LGQSGFALSGGQQQR |
| | | | NIDQQNSAAALSAEK |
| | | | IADVTAFFNAK |
| | | | ATDEALQIVPR |
| | | | GPLSPTLPSLAYLVYEFSR |
| phosphate permease PstA | Tery_3539 | TCCM_0018.00000080 | ATDEALQIVPR |
| | | | GPLSPTLPSLAYLVYEFSR |
| phosphate permease PstC | Tery_3538 | TCCM_0018.00000070 | VLIPAAFSGIVGGVMLGLGR |
| | | | AMGETMAVTMLIGNANSIK |
| SphX periplasmic binding | Tery_3534 | TCCM_0018.00000040 | GAIGYIEFGFAK |
| | | | INLVGAGASFPAPLYQR |
| | | | NSGFEVQVDYQSVGSGAGIER |
| PstS periplasmic binding less phosphate responsive | Tery_3537 | TCCM_0018.00000050 | EVYVDILLGNIK NDGVTAQITQTEGAIGYVEYGYAK |
| | | | VSPELGYIPLPDNVR YIEPTFESAEATLGAVALPENLR |
| flavodoxin | | TCCM_0640.00000010 | IGLFLGTTTGK |
| | | | FVGLALDDDNQAELTDER |

**Table S8.** Literature values for IdiA as a biomarker of Fe stress

| Study | Fe addition replete | Fe addition deplete | IdiA fold change |
|---|---|---|---|
| Webb 2001 | 50nM | 0nM | 1.54 |
| Walworth 2016 | 250nM | 10nM | 1.07 at 380 pCO2, 1.31 at 780 pCO2 |
| Snow 2015 | 120nM | 0nM | 2.38 |

*Note that while the cells in Walworth, et al., 2016 were clearly Fe-limited according to growth rate measurements, they had access to 10nM total iron as opposed to 0nM total iron for the other experiments. This may explain the discrepancy in the IdiA fold change values.

**Table S9.** Literature values for SphX as a biomarker of P stress

| Study | P addition replete | P addition deplete | SphX fold change |
|---|---|---|---|
| Walworth 2016 | 0.25uM | 10uM | 2.64 at 380 pCO2, 3.10 at 780 pCO2 |
| Frischkorn 2019 | 0uM | 50uM | 2.62 |

**Response to reviewer #2 – response text reproduced here with formatting for ease of reading**

**General overview**

*The authors present a metaproteomic study of field-collected Trichodesmium colonies, focused on phosphate and iron stress markers, and complement that study with a membrane crowding model, which I think is a nice approach to try and understand the observed co-limitation patterns for iron and phosphate. The study is comprehensive in that samples from multiple cruises and years are used; with all but one station (HOT) located in the Atlantic Ocean. Increasing the knowledge of nutrient limitation in natural Trichodesmium populations is certainly of interest, given that it seems to be connected to aggregation of Trichodesmium in some way, either directly or through a general C1 stress response. While the study as such is valuable and should be published, I have a few major remarks that I think should be addressed before it is ready.*

We thank the reviewer for their thoughtful comments on our manuscript and discuss changes below in red. Updates to the text are also provided and changes are highlighted in green.

**Major Remarks**

*1. The whole conclusion of co-occurring phosphate and iron stress relies on the assumption that protein abundances of IdiA and SphX are good proxies for iron or phosphate limitation, respectively. The authors do cite the relevant literature that showed upregulation of the respective markers under the corresponding nutrient stress. What I am missing is information on which fold-changes in protein abundance were measured in the cited studies under the respective nutrient limiting conditions. For example, what are the base levels of IdiA and SphX protein in the cell? If there is three times more SphX than IdiA, such as in Fig 3 for some of the Tricolim samples, does that really indicate co-limitation, or does that just reflect the base level of IdiA? For example, Snow et al, 2015 (Fig. 4) only report a two-fold change for IdiA from ~100 fmol/ug to ~200 fmol/ug under iron stress. I suggest presenting the evidence for IdiA and SphX being markers for the respective stresses clearly in a table, including the type of experiment (culture or field), absolute quantifications if stated, and fold-changes measured. I also suggest then being a little more careful in wording throughout the paper, and differentiating better how the results from different stations could be interpreted.*

The reviewer asks us to address how our results compare with previous reports of IdiA and SphX abundance during nutrient limitation. Per the reviewer, we provide new supplementary tables (Table S8 and S9) containing fold-change values from the literature for IdiA and SphX as during nutrient limitation. The biomarkers increased in abundance during nutrient depletion, however the magnitude of the response varied. This is likely due to experimental differences such as the analytical methodology (Western blots versus LC-MS), nutrient concentrations or growth rate of the culture being examined. For instance, IdiA responded less strongly to iron stress in the Walworth et al., 2016

experiments compared to the Webb et al., 2001 and Snow et al., 2015 experiments, and we hypothesize this is because the Fe-depleted condition had 10 nM added iron as opposed to 0 nM iron in Webb et al. 2001 and Snow et al. 2015. We agree with the reviewer that applying a quantitative framework to this data would be valuable once the necessary data becomes available and now note this in the text.

Based on the consistent responsiveness of IdiA and SphX to nutrient limitation in the laboratory, we concluded that the high relative abundance of these biomarkers was indicative of nutrient stress. The reasoning is that the cells were clearly devoting a large fraction of their proteome to Fe and P uptake, likely at the expense of other nutrient uptake systems such as for organic nitrogen as is discussed later in the text. The reviewer's point that there may be basal expression of IdiA and SphX in replete cells is well taken. Because we report relative abundance data, we cannot directly compare our results to those of other researchers who took different quantitative approaches. However, the possibility of basal expression should be clarified in the text. Carefully calibrated datasets relating IdiA and SphX protein abundance to nutrient limited growth rates, while outside the scope of this paper, would be valuable in facilitating interpretation of this data. Based on the reviewer's comments we have updated Section 3.2 below to clarify the assumptions and caveats in our interpretation, and to highlight the need for calibrated biomarker studies.

**3.2 *Trichodesmium* is simultaneously iron and phosphate stressed throughout its habitat**

A surprising emergent observation from the *Trichodesmium* metaproteomes was the co-occurrence of the iron (IdiA) and phosphate (SphX) stress biomarkers across the samples. The ubiquitous and highly abundant presence of these proteins relative to total protein implied that co-stress may be the norm rather than the exception for *Trichodesmium* colonies in the field, particularly in the North Atlantic. Even though low-level basal expression of IdiA and SphX has been observed, it was clear that the colonies were devoting a large fraction of their cellular resources to Fe and P uptake, respectively (see Tables S8 and S9) (Webb et al., 2001, Webb et al., 2007, Chappell et al., 2010, Orchard et al., 2010, Snow et al., 2015, Walworth et al., 2016, Frischkorn et al., 2019). This, combined with the responsiveness of IdiA and SphX to nutrient availability in *Trichodesmium* filaments in the laboratory, indicated that co-stress was occurring.
Interestingly, biomarker abundance was not necessarily associated with nutrient concentrations in the surface ocean, suggesting that the colonies were experiencing stress despite variation in nutrient availability (Figure 3 C-D). SphX abundance varied up to 7.5 fold and were negatively associated with dissolved phosphate concentrations, though analytical differences across the field expeditions may have forced this relationship (Figure S2).  Oceanographically, SphX was most abundant in the P-deplete, summer-stratified North Atlantic gyre (JC150 expedition) compared with winter waters near the Amazon river plume (Tricolim expedition) or at station ALOHA, where phosphate concentrations were greater (Hynes et al., 2009; Sañudo-Wilhelmy et al., 2001; Wu et al., 2000). IdiA varied up to 8 fold but there was no observable relationship with dFe concentrations at the surface. Instead, IdiA may be responsive to other factors such as the

varying iron requirements of the populations/species examined. It should be highlighted that in this study only *Trichodesmium* colonies were considered, so factors such as colony size may affect iron availability and biomarker expression. Additionally, because the surface ocean iron inventory is low, transient inputs such as from the Sahara desert can dramatically impact iron availability on short time scales, and the time scale of these inputs relative to changes in biomarker abundance is not well understood (Kunde et al., 2019). Carefully calibrated datasets relating IdiA and SphX abundance to nutrient-limited growth rates of *Trichodesmium* in both the filamentous and colonial forms would facilitate further interpretation of this data.

**Table S8.** Literature values for IdiA as a biomarker of Fe stress

| Study | Fe addition replete | Fe addition deplete | IdiA fold change |
|---|---|---|---|
| Webb 2001 | 50nM | 0nM | 1.54 |
| Walworth 2016 | 250nM | 10nM | 1.07 at 380 pCO2, 1.31 at 780 pCO2 |
| Snow 2015 | 120nM | 0nM | 2.38 |

*Note that while the cells in Walworth, et al., 2016 were clearly Fe-limited according to growth rate measurements, they had access to 10nM total iron as opposed to 0nM total iron for the other experiments. This may explain the discrepancy in the IdiA fold change values.

**Table S9.** Literature values for SphX as a biomarker of P stress

| Study | P addition replete | P addition deplete | SphX fold change |
|---|---|---|---|
| Walworth 2016 | 0.25uM | 10uM | 2.64 at 380 pCO2, 3.10 at 780 pCO2 |
| Frischkorn 2019 | 0uM | 50uM | 2.62 |

*2. Figure 3, and the corresponding Figure S2 are nice and the basis for some important claims being made in part 3.2. of this manuscript. However, these claims should be supported with the necessary statistics, and it would help if Fig S2 was not in the Supplement, but presented together. For example, in line 210 ff, the authors claim that a) "Biomarkers for iron (IdiA) and phosphate (SphX) stress were highly abundant and positively associated with surface Fe or P concentrations" and b) "IdiA varied up to 8 fold, and increased moving West to East across the JC150 transect, consistent with an observed decrease in dFe concentrations". For a) I think the authors mean "negatively", not "positively", correlated. And while I believe this correlation for SphX, it is not obvious for IdiA. For b) I cannot see increasing protein abundance from west Please prove this statistically before claiming it.*

First, we thank the reviewer for correcting our mistake in line 210 – we did indeed mean to write "negatively." For clarity, we have moved the panels in Figure S2 alongside Figure 3 in the main text. Figure S2 now provides scatter plots of IdiA and SphX versus dFe and phosphate concentrations in the surface ocean (see updated figures section at the end of this document). There was a statistically observable relationship between SphX and dissolved phosphate, however the relationship may be forced by the different analytical approaches used on the JC150 versus Tricolim expeditions as is noted in the text below. By contrast there was no statistically observable relationship between IdiA and dFe, even though laboratory experiments clearly indicated that IdiA was a good biomarker of Fe stress in *Trichodesmium*. There are many factors that could influence this association, particularly iron speciation, changes in iron quotas, and factors such as colony size, which are not controlled in the field. We thank the reviewer for calling this point to our attention as it provides an opportunity to discuss these points in the updated Section 3.2 (see Major Remarks #1).

*3. Given that only 1 sampling station is NOT in the Atlantic, please remove all claims that generalize the findings, e.g. "co-stress is the norm rather than the exception" (l. 18) → add "in the Atlantic", if wanting to keep this. Or in line 60f: "simultaneously Fe and P stressed throughout the worlds oceans" – this statement cannot be made with just one station outside the Atlantic.*

Change accepted.

**Specific Comments**

**Abstract**

*The abstract is missing some specificity.*

*line 19: nitrogenase was most abundant – compared to what? Please rephrase: more abundant than under . . line 22: is confronted by the biophysical limits – when? Under which conditions is it confronted by this? line 24f: be more specific. The last sentence is true for any microbe.*

We agree with these points and have updated the text accordingly. On line 24f, we wished to highlight the importance of considering multiple nutrients for *Trichodesmium* specifically, given the historical emphasis on either Fe or P stress.

**Abstract.** *Trichodesmium* is a globally important marine microbe that provides fixed nitrogen (N) to otherwise N limited ecosystems. In nature, nitrogen fixation is likely regulated by iron or phosphate availability, but the extent and interaction of these controls are unclear. From metaproteomics analyses using established protein biomarkers for iron and phosphate stress, we found that co-stress is the norm rather than the exception for field *Trichodesmium* colonies. Counter-intuitively, the nitrogenase enzyme was ==more abundant under co-stress than under single nutrient stress==, consistent with the idea that *Trichodesmium* has a specific physiological state under nutrient co-stress, ==as opposed to==

==single nutrient stress==. Organic nitrogen uptake was observed to occur simultaneously with nitrogen fixation. Quantification of the phosphate ABC transporter PstA combined with a cellular model of nutrient uptake suggested that *Trichodesmium* is ==generally== confronted by the biophysical limits of membrane space and diffusion rates for iron and phosphate acquisition ==in the field==. Colony formation may benefit nutrient acquisition from particulate and organic nutrient sources, alleviating these pressures. The results ==highlight== that to predict the behavior of *Trichodesmium*, ==both Fe and P stress must be evaluated and understood simultaneously.==

**Introduction**

*Line 36: colloquialism*

Change accepted.

*Line 58: add . . .Pho box, a regulatory DNA sequence, which is necessary. . .*

Change accepted

*Line 65f: Fe and P stress were positively associated – only as co-stress? If yes, say so.*

They were associated both individually and under co-stressed.

*Also say how Fe, P, and N statuses are closely linked.*

We suggest these are linked via a currently unknown regulatory network; change accepted.

**Methods**

*Line 119: what does that mean? Which precursors, of what? Was every protein normalized to the top 3 precursor intensities? Make this clear also to a reader who is not familiar with the specifics of proteomics analysis.*

Relative abundance was measured by averaging the peptide precursor/MS1 intensities for the 3 most abundant peptides in the protein, then normalizing this value to the total precursor intensity. Text has been updated:

Raw spectra were searched with the Sequest algorithm using a custom-built genomic database (Eng, Fischer, Grossmann, and MacCoss, 2008). The genomic database consisted of a publically available *Trichodesmium* community metagenome available on the JGI IMG platform (IMG ID 2821474806), as well as the entire contents of the CyanoGEBA project genomes (Shih et al., 2013). Protein annotations were derived from the original metagenomes. SequestHT mass tolerances were set at +/- 10ppm (parent) and +/- 0.8 Dalton (fragment). Cysteine modification of +57.022 and methionine modification of +16 were included. Protein identifications were made with Peptide

Prophet in Scaffold (Proteome Software) at the 95% protein and peptide identification levels. Relative abundance was measured by averaging the precursor intensity (area under the MS1 peak) of the top 3 most abundant peptides in each protein, then normalizing this value to total precursor ion intensity. Normalization and global false discovery rate (FDR) calculations, which were 0.1% at the peptide level and 1.2% at the protein level, were performed in Scaffold (Proteome Software). FDR was calculated by Scaffold using the probabilistic method by summing the assigned protein or peptide probabilities and dividing by the maximum probability (100%) for each. The mass spectrometry proteomics data have been deposited to the ProteomeXchange Consortium via the PRIDE partner repository with the dataset identifier PXD016225 and 10.6019/PXD016225 (Perez-Riverol et al., 2019). Statistical tests of relationships between proteins were conducted with the scipy stats package (https://docs.scipy.org/doc/scipy/reference/stats.html) using linear Pearson tests when the relationship appeared to be linear and a Spearman rank order test when this was not the case.

*Line 120: How is the FDR defined? What does "0.1% peptide" and "1.2% protein" mean?*

Text updated (see above paragraph):

FDR was calculated by Scaffold using the probabilistic method by summing the assigned protein probabilities and dividing by the maximum probability (100%) for each. Different FDRs can be assigned for peptides versus proteins depending on which probabilities are used for the calculation.

*Line 128: Which peptides were selected?*

We have added to the supplemental Tables S6 and S7 which describe the peptides selected for quantitation

**Results and Discussion**

*Line 178: change "most" to "all but one"*

Change accepted

*Line 232: please rephrase. What exactly is common in marine bacteria. For sure, all bacteria have regulatory networks.*

Change accepted. A recent review of regulatory genes found that regulatory networks may be particularly more abundant in marine organisms (Held et al., 2019).

This indicates that the cell's N, P and Fe statuses are linked, perhaps involving a regulatory network which are particularly common in marine bacteria (Figure 5) (Held et al., 2019).

*Line 255ff: skip this justification sentence*

With respect we prefer to leave this sentence in because exploratory metaproteomics is not yet widely used as an analytical tool in oceanography. We hope this study and others like it will encourage its adoption. We do welcome editorial advice or further discussion on this point.

*Section 3.4. Throughout this section, I think the use of the term ligand is not the norm. For ABC transporters, the word "ligand" is typically used for whatever binds to and is transported by the transporter. The part of the transporter binding the substrate is usually called "ligand-binding protein".*

We agree with the reviewer that "ligand" is often used to describe a chemical compound, for instance a siderophore, which can be transported by an ABC transporter. However specifically in the uptake kinetics literature "ligand" is used to describe the ABC transporter itself (i.e. the protein that binds the nutrient). To avoid confusion we have updated the text to use the word "transporter" or "protein" instead.

*Line 260: change to ". . .required for both iron and phosphate uptake"*

Change accepted

*Line 305ff: rewrite sentence. Hard to understand.*

Updated, and hopefully clearer now!

==For a given surface area: volume quotient, we define nutrient limitation to be caused by either membrane crowding or diffusion limitation depending on which model calculated a higher minimum nutrient concentration.==

*Line 316ff on cylinders: Shouldn't the Trichodesmium filament, instead of a single Trichodesmium cell, be considered for these models? The effective cell surface of a Trichodesmium cell is reduced by its contact to the neighboring cells.*

The reviewer is correct that membrane limitation would be exacerbated for cells living in filaments, as the surface area exposed to the surrounding environment would be reduced. We considered modeling filamentous cells but decided to consider only single cells for clarity since this is the most conservative scenario (i.e. the one in which *Trichodesmium* would theoretically have the most exposure to the environment and be the least limited). We have added a sentence to the discussion of the model highlighting the focus of the model (single cells) but mentioning that filamentous cells would have lower surface area.

While this model may be directly applicable to some $N_2$-fixing cyanobacteria such as Groups B and C, which have roughly spherical cells, *Trichodesmium* cells are not spheres but rather roughly cylindrical (Hynes et al., 2012). Thus, we repeated the model calculations for cylinders with varying radii (r) and heights (2r or 10r) based on previous

estimates of *Trichodesmium* cell sizes (Bergman et al., 2013; Hynes et al., 2012). Cylinders have lower surface area: volume quotient than spheres of similar sizes. In addition, the rate constant ($k_D$) for diffusion, which is a function of cell geometry, is greater. This increases the slope of the diffusion limitation line such that membrane crowding is important across a greater range of cell sizes (Figure 7c-d). *Trichodesmium* cell sizes vary in nature, for instance the cylinder height can be elongated, improving the surface area: volume quotient. However, the impact of cell elongation to radius r and height 10r on both diffusion limitation and membrane crowding is subtle (Figure 7e-f). Furthermore, though not explicitly considered here, cylindrical cells living in filaments would have reduced surface area available for nutrient uptake. Thus, we conclude that in certain scenarios, lack of membrane space could hypothetically limit Fe and perhaps P acquisition by *Trichodesmium*.

*Line 361: Reference missing for mucus production being a "hallmark of Trichodesmium colony formation"*

Citation added (Eichner et al., 2019)

*Line 362f: If mucus acts as a diffusive barrier, it also does the opposite of "protecting them [the cells] from oxygen", namely preventing O2 to diffuse out of the cells during photosynthesis, which was also shown in Eichner et al, 2019.*

Note added to this effect:

A key hallmark of *Trichodesmium* colony formation is production of mucus, which can capture particulate matter and concentrate it within the colony (Eichner et al., 2019). In addition to particle entrainment, the mucus layer can benefit cells by protecting them from oxygen and/or concentrating oxygen during photosynthesis, facilitating epibiont associations, regulating buoyancy, defending against grazers and helping to "stick" trichomes together (Eichner et al., 2019; Lee et al., 2017; Sheridan, 2002). However, these benefits come at a cost because the mucus layer hinders diffusion to the cell surface (Figure 9), reducing contact with the surrounding seawater. Despite this, the benefits of colony formation seem to outweigh the costs, since *Trichodesmium* forms colonies in the field, particularly under stress (Bergman et al., 2013; Capone et al., 1997; Hynes et al., 2012).

*Line 384: Which specific regulatory systems should be characterized? What do you mean by chemical phases?*

We don't know yet which regulatory systems should be examined! For a review of marine regulatory systems and their often unknown functions see Held et al. 2019. These results suggest that one or more regulatory networks may control Fe, P, and N status in tandem with one another in *Trichodesmium* cells, and we hope this work will stimulate future research on this topic. By chemical phases we meant dissolved versus particulate nutrient sources, since *Trichodesmium* is known to use both – we've clarified this now:

Future studies should aim to characterize the specific regulatory systems, chemical species and phases (i.e. dissolved versus particulate nutrient sources), and symbiotic interactions that underlie *Trichodesmium's* unique behavior and lifestyle.

**Figures**

*Figure 1: Please use the same numbers on the figure and the legend, or at least also add the figure numbering top the legend.*

Figure 1 has been updated (see updated figures below).

*Figure 2: Please increase the font size on the legend, and add a legend name like "# of times a protein appeared in the same cluster" – consider changing the legend to a percentage. Please also say in the caption what the color legend shows.*

Figure 2 and its caption have been updated

*Figure 3: Please state in the caption how the protein abundance values were normalized.*

Figure 3 caption has been updated:

"Figure 3. (A) Relative abundance of iron stress protein IdiA (A) and phosphate stress protein SphX (B). IdiA and SphX were among the most abundant proteins in the entire dataset. Error bars are one standard deviation on the mean when multiple samples were available. Dashed lines represent average values across the dataset. Proteins abundances were normalized such that the total MS1 peak area across the entire proteome was the same for each sample. (C) Relative abundance of IdiA (orange) and SphX (blue) overlaid on the sampling locations."

*Figure 4: Please adjust font sized throughout panels. How were the dashed lines in c and d defined? Based on what do they denote Fe- or P-stress? And why are they different in c and d?*

Figure 4 font sizes have been updated. The dashed lines in (C) and (D) were drawn by hand to help the reader to visually understand the intersection of Fe and P stress. However, given the important discussion about IdiA and SphX abundances raised by the reviewer, we can see how this might be misleading and have removed the dotted lines.

*Figure 5: Does not necessarily need to be a figure if wanting to save space.*

We welcome editorial advice on this but would advocate for including the figure as we think it illustrates the discussion in section 3.3, particularly for visual learners.

---

## Author Response (AR2)

**Authors Response to Associate Editor report**

We thank the editor for their careful consideration of our manuscript. All of the technical corrections suggested by the editor have been made in the latest version.

Additionally, for clarification we make the following wording changes to section 3.2:

Additionally, because the surface ocean iron inventory was low, transient dust inputs such as from the Sahara desert could dramatically impact iron availability on the short time scales of particle deposition, sinking, solubilization, and iron uptake. The time scale of these processes relative to changes in biomarker abundance is not well understood (Kunde et al., 2019). Carefully calibrated laboratory datasets relating IdiA and SphX abundance to nutrient-limited growth rates of *Trichodesmium* in both the filamentous and colonial forms would facilitate quantitative interpretation of this data.